# The Effectiveness of Local Governments’ Policies in Response to Climate Change: An Evaluation of Structure Planning in Arden, Melbourne

**DOI:** 10.3390/ijerph20032469

**Published:** 2023-01-30

**Authors:** Jiawen He, Xinting Xie, Fengchen Luo, Yanfen Zhong, Ting Wang

**Affiliations:** 1Melbourne School of Design, The University of Melbourne, Melbourne, VIC 3010, Australia; 2School of Architecture and Urban Planning, Beijing University of Civil Engineering and Architecture, Beijing 100044, China

**Keywords:** structure plan, evaluation, climate change, liveability, Arden, Melbourne

## Abstract

It is widely acknowledged that climate change has caused serious environmental issues, including drought, bushfires, floods, and heatwaves, and urban sustainability is currently seriously threatened as a result. Arden is one of the key urban regeneration areas set to experience dramatic residential changes under Melbourne’s development blueprint within the next 20 years. The Arden Structure Plan (2022) outlines specific implementation steps but does not go into detail about the strategies and tactics used to address climate change and urban sustainability. Therefore, there are still problems with the plan, including a lack of information and time-bound development targets, ambiguous public engagement, little focus on urban crime, and insufficient climate change adaptation measures. The plan also considers affordable housing, a mixed-use development pattern that will significantly decrease environmental harm, and active transportation options, primarily walking and bicycling. Considering climate change, this plan will make Arden a suitable location for population growth. This paper aims to evaluate the Arden Structure Plan and make recommendations on how to improve the plan’s urban sustainability and climate change considerations. Furthermore, it provides guidance on whether Arden is a suitable location for Melbourne’s population growth in light of the climate change impacts anticipated to occur by 2100.

## 1. Introduction

Structure planning is a type of spatial planning and is a part of the urban planning practices in the United Kingdom and Australia [1]. In Australia, structure plans are commonly prepared at the subregional, district, and local levels under strategic planning. Structure plans set the direction of development in urban sub-regions and play an important role in the implementation and effectiveness of strategic policy objectives. They also make provisions for changing community needs and set out how these needs should be managed by the local government or councils. They guide major changes in land use, transportation integration, built form, public space, and cultural transmission, working together to achieve the economic, social, and environmental objectives of the area [2,3].

In the last hundred years, the world has experienced significant changes, mainly characterised by warming, with widespread and far-reaching effects [4,5,6]. In Australia, particularly in metropolitan areas such as Sydney and Melbourne, the pressures of population growth and the effects of climate change compound the problems of environmental degradation, high resource consumption, and the loss of development space [7,8]. The United Nations Environment Programme (2021) claims that, as emission pledges between nations increase in size and complexity, climate change becomes a more serious issue [9]. Additionally, national governments are unable to make significant contributions towards effectively reducing greenhouse gas emissions (for example, by repealing Australia’s carbon tax). As a result, local governments play a crucial role in directing sustainable development. As of 2022, the population of Melbourne is about 5 million, and the population density in the city is about 500 residents per square kilometre. In fact, around 75% of the population of Victoria lives in Melbourne. Melbourne is not only a populated city, but it is also expanding quickly. Melbourne is expected to overtake Sydney to become Australia’s largest city in about 25 years if it continues to grow at this rate [10]. Structure plans must take on additional functions not only to ensure the economic and social vitality of the area, but also to develop a range of tools to help local governments create and manage their plans to make them more resilient to climate change and increase the liveability of these regions [11].

Given that planning decisions have decades-long impacts on both the natural and built environments, which are sometimes irreversible [12], conducting programme assessments is a critical step in ensuring that these plans produce the desired results [13,14]. Planning assessments can help to increase the accountability of local governments and public institutions and inform decision makers on ways to address policy issues [15], ensuring that the planning is serious and scientific [16]; in addition, they also strengthen public confidence in planning decisions [17,18]. Scholars have developed analytical methods and conducted empirical studies to assess the quality of plannings [19,20]. The question of whether planning assessments should focus on ‘ex ante (or a priori)’, ‘ongoing’, or ‘ex post’ has been widely discussed by scholars in various countries [21,22,23,24,25]. Although quantitative indicators play a pivotal role in planning, the joint assessment of qualitative-description methods is comprehensive, especially for those planning implementation goals that cannot be directly quantified, in which the description of qualitative indicators is essential [26]. In the 1980s, the Canadian scholar Hok-Lin Leung proposed the S-CAD (subjectivity, consistency, adequacy, and dependency) public policy evaluation method, which helps policy-making participants develop and analyse their own policy recommendations as well as evaluate and resolve their differences with other participants [27,28,29]. Many governments have also established planning-evaluation systems and have applied planning evaluation methods in practical work. The UK represents the Western country with the most complete planning evaluation system, which was eventually adopted by the EU to form a regional planning policy evaluation system under the EU framework [30]. Portland’s Comprehensive Plan Assessment plan, based on an analysis of the implementation environment and urban development trends of Portland’s 1980 master plan, focuses on assessing the timelines of the implementation of the plan and provides a systematic assessment of the master plan in seven areas: economic development, environmental protection, housing planning, infrastructure planning, sustainable development, transportation, and urban form [31]. Hong Kong’s Territorial Development Strategy Review focuses on assessing the forward-looking nature of planning, particularly in terms of assessing Hong Kong’s competitiveness in the global urban system [32,33].

Sustainable development and combatting climate change are currently some of the most topical issues in urban research and are among the core objectives of urban development. Planning assessment methodologies that address sustainable urban development, climate change, and the maintenance of infrastructure support and ecological and green spaces have been developed, a few examples being the Urban Sustainability Assessment Framework and the Climate Change Assessment Framework. Holden et al. suggested that urban sustainability should be based on three imperatives: satisfying human needs, ensuring social equity, and respecting environmental limits, identified by six key themes [34]. This emphasises the primary objective of planning and proposes quantifiable indicators, which makes this definition more forward-looking and feasible at both the state level and municipality level. Although the neighbourhood is considered the most appropriate spatial scale for improving sustainable built environments [35], bringing sustainability to small neighbourhoods has been acknowledged as a challenge for developers and local municipalities because of a lack of institutional and financial capacities. Compared with larger communities, smaller neighbourhoods are limited in their ability to adopt sustainable practices. Neighbourhood sustainability assessment (NSA) tools have been introduced to measure the effectiveness of sustainable development at this spatial scale [36], including LEED (Leadership in Energy and Environmental Design), BREEAM (Building Research Establishment Environmental Assessment Method), and ITACA (Italian Institute for Innovation and Transparency in Procurement and Environmental Compatibility). Haider et al. developed a sustainability assessment framework for small urban neighbourhoods based on analyses using several NSA tools [37]. These NSA tools are considered reliable in assessing the potential for sustainable development in precincts and achieving sustainability goals [38]. In accordance with this reliability, sustainability assessment tools can also be used to assess the degree of consideration of sustainable development in specific strategic plan documents. The consideration of climate change in municipalities’ plans provides a crucial opportunity to address this global issue, as cities are recognised as major drivers in addressing climate change actions. Larsen and Kørnøv presented three overall approaches to address climate change: mitigation, adaptation, and baseline adaptation [39]. Grafakos et al. indicated the significance of integrating mitigation and adaptation into the global transition toward responding to climate change [40]. This can help avoid conflict between different policies and bring far-reaching benefits for enhancing collaboration. Mitigation is considered a global-scale issue. However, with further research and the advancement of strategic climate change plans, there is a growing awareness of the impact of bottom-up behaviour on mitigation action. Thus, it is important to consider mitigation at both the global level and local municipal level. In addition, considering the roles of adaptation and integration is also crucial in creating climate change assessment frameworks. In order to assess the effectiveness and advancement of these plans for addressing climate change, Hurlimann et al. introduced an assessment focusing on the actions of mitigation, adaptation, and integration [41].

The quality of a plan is a key determinant of its outcome, and a high-quality plan is more likely to produce positive outcomes [42]. The five million residents of Melbourne deal with serious environmental issues, including drought, bushfires, floods, and heatwaves [43], especially in Arden, which experiences severe flooding. Therefore, it is necessary to critically evaluate the structure plan for the Melbourne urban renewal area of Arden. The government is looking for suggestions on how to better take climate change and urban sustainability into account in the plan and whether the urban renewal project should move forwards. The primary objective of this study was to evaluate the Arden Structure Plan and provide suggestions on how to improve the plan’s urban sustainability and climate change considerations. Secondly, we aimed to offer guidance on whether Arden is a suitable location for Melbourne’s population growth considering the impacts of climate change anticipated to occur by 2100.

## 2. Methodology

### 2.1. Study Area

Arden—situated between Macaulay Road, the Upfield railway line, Moonee Ponds Creek, and Dryburgh Street—is an urban regeneration area in North Melbourne located approximately 2 km from the Melbourne CBD (Figure 1) that comprises 44.6 hectares of industrial land. Arden is situated in a floodplain and is vulnerable to flooding. According to the Victorian Planning Authority (2022), Arden is expected to provide approximately 34,000 jobs and accommodate approximately 15,000 residents over the next 15 to 30 years and will become an employment and transport precinct and a new community for inner Melbourne [44].

The Arden Structure Plan and Planning Scheme Amendment C407 was approved by the Minister for Planning and gazetted on 28 July 2022 under Amendment C407melb of the Melbourne Planning Scheme. Thus, Arden will seize the opportunity to set the standard for best practices in innovation and leadership for sustainable urban renewal, including a commitment to achieve net zero emissions for the entire precinct by 2040 in accordance with the Melbourne City Council’s policy. The Arden Structure Plan aligns with several local plans proposed by the City of Melbourne.

### 2.2. Method

#### 2.2.1. Assessment Framework

As previously mentioned, there has been a growing interest among planning scholars in evaluating the quality of a range of plans, especially those focusing on natural hazards [45], ecosystem management [46], climate change [47], and sustainable development [48]. However, there is a lack of common criteria and frameworks for assessing the quality of these plans. Kaiser et al. highlighted three core characteristics that constitute a high-quality framework: a strong factual basis, clear goals, and appropriate policies [24]. These characteristics can be used to assess the quality of a plan in regard to its strategic direction and statement via the quantitative method [49]. Berke et al. expanded these three characteristics to eight characteristics, adding implementation, monitoring and evaluation, inter-organisational coordination, participation, and plan organisation and presentation [24]. Elgendawy et al. adopted an AAA approach based on three critical components: awareness (targets and analysis), analysis, and action (implementation and monitoring to assess plan quality) [49].

In order to assess the appropriateness of the sustainability and climate change considerations outlined in Arden’s plan and the feasibility of its actions, this study conducted a plan quality evaluation by assessing the content of the plan with respect to sustainability and climate change. Here, the AAA approach was adapted to form a new plan quality assessment protocol composed of 14 indicators designed to focus on these two themes. The overall framework is illustrated in Figure 2. It covers the three core factors of awareness, analysis, and action. For the awareness factor, there are four indicators: context, concept, impacts, and targets. For the analysis factor, we adapted an NSA framework [37] and a climate change assessment framework [41] to measure the consideration of these two themes and their specific quantitative indicators in the plan. Although this NSA framework is focused on assessing the current state of sustainability within a site, the criteria were used as a reference in this study to measure whether they were considered and developed in the Arden Structure Plan. This NSA framework has six objectives: resource efficiency, economic prosperity, social well-being, public mobility and accessibility, environmental quality, and natural-land protection. These six objectives are based on three moral imperatives—satisfying human needs, ensuring social equity, and respecting environmental limits—and they are adapted from the six themes identified by Holden et al. [34]. In addition, the climate change assessment framework used here, adopted from Hurlimann et al. [41], consists of three parts: mitigation, adaptation, and integration. It focuses on the actions to address climate change in the Arden Structure Plan and highlights considerations at the state level and on a smaller scale. The actions component of our framework identifies appropriate measures and monitoring strategies for demonstrating how the local government will achieve sustainable urban development and reduce the trend of climate change, which include implementation, monitoring, and evaluation as well as inter-organisational coordination and participation. In this study, we combined both qualitative and quantitative methods to develop the plan quality assessment framework. The content of the Arden Structure Plan was analysed using a descriptive evaluation and scored according to a scoring protocol. Here, we present an overall view and a conclusion of the plan’s quality, focusing on the outcomes of the urban sustainability assessment and climate change assessment, and discuss the shortcomings and potential of the proposed development in Arden.

#### 2.2.2. Fieldwork and Scoring Protocol

This study included 15 Melbourne- and Beijing-based urban planning and urban geography scholars, 10 of which held PhDs, including one professor, three associate professors, six assistant professors, and four current master’s students. In order to ensure an objective understanding and real perception of Arden’s current situation, and to increase the accuracy and reliability of the scoring results, four members of the study carried out fieldwork on 16 September 2022; their route can be seen in Figure 3. Point A is the North Melbourne train station, which is currently the closest train station to Arden. The streets on this route will provide quick and convenient walking and cycling connections between key spaces such as the Macaulay, Arden, and North Melbourne train stations, as well as open spaces within and outside of Arden. Point B is the new Arden station, due to open in 2025, which will accelerate Arden’s emergence as the centre of its district and assist in achieving the target of at least 60% public transport, 30% walking and cycling, and no more than 10% private vehicles. Points C, D, and E are the North Melbourne Recreation Reserve, underground storage tanks, and temporary parking, respectively. Water flow from Melbourne’s inner northwest has an impact on the Moonee Ponds Creek (Point F). The town’s current drainage system is not designed to handle this amount of stormwater, resulting in dangerous flood depths. According to Figure 4, taken from the region’s drought vision framework, a sizable portion of land will be at risk of flooding, which will undoubtedly become more severe if sea levels rise by 2.7 m as described in the worst-case scenario [50]. Point G and Point H are outside the reach of the Arden Structure Plan; however, it is worth noting that the residential houses in these areas are also vulnerable to flooding.

The scholars scored the Arden Structure Plan with reference [51,52] to the scoring methods used in previous studies. An evaluation of the quality of the Arden Structure Plan was conducted by scoring the indicators on a scale between 0 and 2 [49]. A score of ‘0’ was given if there was no evidence or description of the indicator throughout the plan. An indicator was given a score of ‘1’ if it was inferred or mentioned but not in detail, while an indicator was given a score of ‘2’ when it was fully mentioned and discussed in detail. Each factor (AAA) was equally important and had the same weight. This scoring protocol was implemented with comments on each indicator to justify the coding decision. The evaluation protocol contained 14 indicators with a maximum score of 28 points. These indicators were divided according to the AAA approach as follows: awareness (4/14), analysis (6/14), and action (4/14). Scoring was performed for each category, with a maximum of 8 points for awareness, 12 points for analysis, and 8 points for action. In addition, an assistant professor from the University of Melbourne was invited to validate the scoring protocol to ensure professionalism and reduce the influence of subjective factors by averaging the results.

## 3. Results

### 3.1. Overview

Figure 5 illustrates the quantitative results of the quality evaluation, including the mean scores for the three core factors: awareness, analysis, and action. There were significant differences in the scores for the different core factors. Higher scores indicate that the factor is more fully described in the Arden Structure Plan. The analysis factor had the lowest score of 1.13, while the action factor had the highest score of 1.5. This shows that the Arden Structure Plan (2022) has clear implementation steps, but its analysis needs to be more specific and systematic. Each core factor will be discussed in some depth in other parts of this section. A complete list of indicators and their evaluations are presented in Appendix A.

### 3.2. Quality of Awareness

The mean score of the awareness factor, which contained four indicators, was 1.25. This indicates that the Arden Structure Plan (2022) covers basically all indicators, but the description of most indicators is vague. The plan has clear goals and measures for urban sustainability and climate change, embedding sustainable change as an important chapter by proposing many detailed objectives and strategies. However, the plan does not explicitly describe the context of urban sustainability and climate change. The plan discusses the knowledge, concepts, and impacts of climate change and urban sustainability in general terms. Relevant contexts, such as the heat island effect, are scattered throughout the plan without any systematic description.

### 3.3. Quality of Analysis

The analysis factor, which was composed of urban sustainability and climate change indicators, received a relatively low score. This indicates that the Arden Structure Plan (2022) lacks a detailed description of measures and strategies for urban sustainability and climate change. In addition, Arden is a precinct, not a large city; thus, the plan needs to mention more specific development strategies.

#### 3.3.1. Urban Sustainability

(1)Environmental Limits

Environmental limits relate to three sustainability objectives of natural-land protection, resource efficiency, and environmental quality. The mean score for natural-land protection was ranked lowest among the sustainable objectives, with a mean score of 0.74 (Figure 6). The Arden Structure Plan (2022) focuses on efficient land use to create a new urban structure for Arden while neglecting to protect the land with respect to urban construction. For example, the protection of imperilled species, the preservation of buildable land for green space, etc., are closely related to biodiversity loss, land-system change, and other environmental limits. In addition, the other two sustainability objectives are also not described comprehensively in the plan. For resource efficiency, the plan focuses on recycling resources but has little mention of how to improve the durability of urban resources, such as through the application of technology. With respect to environmental quality, considerations of comfort in both indoor and outdoor environments are not proposed specifically in the plan. For instance, there is a lack of both protective measures against strong winds and standards for distance from the critical floodplain.

(2)Human Needs

The Arden Structure Plan (2022) performed relatively well in terms of human needs when compared with the other two indicators, focusing primarily on the sustainable objectives of public mobility and accessibility, social well-being, and economic prosperity, whose respective mean scores were 1.12, 1.33, and 1.17 (Figure 7). The plan proposes convenience, safety, and accessibility for transportation, encouraging people to walk, cycle, and use public transport to meet their daily needs. At the same time, it facilitates a mix of land uses to create an employment-focused, amenity-rich, and innovative mixed-use region [3]. The plan proposes a series of objectives and strategies aimed at addressing the basic needs of residents. However, the plan fails to address social security, for example the crime rate, and does not have a specific description of public service facilities, such as urban furniture, which are also significant for residents’ living standards. In addition, the plan lacks specific data to support its validity, for example with respect to the specific distance from residential areas to educational and cultural facilities.

(3)Social Equity

Social equity, which includes the sustainable objectives of public mobility and accessibility, resource efficiency, and social well-being, was found to be a weak element of the Arden Structure Plan (2022). Public mobility and accessibility had the lowest score at 0.75 (Figure 8). Holden et al. defined social equity as fair civil rights and an equal distribution of resources (especially for the most disadvantaged groups) [34]. The Arden Structure Plan does not provide a clear description of how the public can share urban public spaces equally, and city residents did not participate in relevant urban designing during the creation of the plan, ignoring their potential rights. The plan also lacks a systematic design of public service facilities for disadvantaged groups; instead, there are only a few scattered strategies, such as parking for disabled individuals. Moreover, though the plan does well to provide a certain percentage of affordable housing for vulnerable groups, it neglects to recognise some different identities, such as LGBTQ individuals and new arrivals. Most of the content of the plan relates to the physical environment and urban construction, and descriptions of social well-being and equity are vague.

#### 3.3.2. Climate Change

(1)Mitigation

The mean score for climate change mitigation was 1.73 (Figure 9). This indicates that the Arden Structure Plan (2022) outlines comprehensive mitigation-related strategies and policies. The plan addresses climate change mitigation and related goals. It also describes climate mitigation strategies at the local level, including housing, mixed-use zoning, transportation, self-contained communities, and pedestrian zones. However, the plan still leaves out some indicators; it does not consider the construction of ecological communities or the control of traffic flow on different scales of roads, which are also significant factors for climate change mitigation.

(2)Adaptation

The Arden Structure Plan (2022) was found to address climate change adaptation with a mean score of 1.2 (Figure 9). In the Arden precinct, adaptation to climate change mainly concentrates on flood management. “Celebrating Water” is one important section of the plan that includes the use of natural and built infrastructure to mitigate the impacts of flooding and heavy rain. However, the plan’s approach to flooding in the plan mainly considers accommodation and protection, and it does not mention the diversion of wastewater. Adaptation focuses on overcoming and resisting the negative impacts of climate change [53]. The strategies described in Arden’s plan are more related to mitigation.

(3)Integration of Adaptation and Mitigation

The average score for the integration of adaptation and mitigation was 1.0 (Figure 9), which indicates that the Arden Structure Plan (2022) is quite vague in its emphasis of these topics. For climate change, the plan mentions adaptation and mitigation measures and strategies in different objectives. However, the plan focuses more on mitigation than adaptation, and both are rather one-sided and are not described systematically.

### 3.4. Quality of Action

The action factor included implementation, monitoring, and the participation of different actors. According to the evaluation results, some of the indicators are mentioned specifically in the plan, while others are described generally. The Arden Structure Plan (2022) clearly proposes development staging, early activation, and infrastructure funding in the section called “Delivery Arden” to ensure its implementation. The plan focuses on connecting with different policy agendas at different levels of government, such as in the Moonee Ponds Creek Strategic Opportunities Plan, to promote their shared contribution to urban construction. However, there are some drawbacks to the plan’s participation mechanism. It only lists the nominated lead agency in the final implementation table and does not describe the reasons why they were asked to participate in the actions. In addition, the responsibilities of stakeholders are not provided in the plan. In this case, it is difficult to perform a rigorous monitoring and evaluation of its implementation.

## 4. Discussion

The Arden Structure Plan was published in July 2022. Although we did not assess further planning options, we did assess the planning issues, planning objectives, and strategies that the plan could address [54,55]; thus, this work is similar to an ex ante evaluation. A considerable effort was made to try to predict whether the planning objectives and solutions outlined in the Arden Structure Plan are scientific and relevant in the context of climate change and sustainable urban development. In this study, we integrated an assessment of urban sustainability and climate change with an assessment of the quality of the plan, as well as constructed an assessment framework that combined the quantitative and qualitative assessments of not only the urban sustainability and climate change elements of the Arden Structure Plan but also the feasibility and public participation of the plan. As Baer pointed out, there is no single standard for planning assessment; the established standards only provide a reference for planning assessment practitioners and are not decisive [56]. It is often possible to interpret them systematically by setting corresponding criteria [57]. For example, we have set up indicators such as an “implementation section” and a “monitoring and evaluation section” with criteria dependent on whether the plan includes separate sections that address what needs to be done to implement the plan or monitor and evaluate the plan, respectively, and other criteria to assess the feasibility of the plan. We additionally set indicators for “stakeholder and public engagement,” where public participation was assessed through asking whether the plan identifies the organisations and stakeholders involved in the plan-making process and identifies the public as part of the plan-making process. This assessment framework draws on relevant indicators from existing references [24,49] and adapts them to the water-sensitive geographical characteristics of Arden’s floodplain, focusing more on the impact of flooding on the area by setting criteria such as the presence of groundwater pumping, the percentage of 1-year, 24-h storm runoff for the entire community, and the community’s distance to the critical floodplain for the next 100–1000 years. This assessment framework is friendly and flexible for planners in water-sensitive areas and can help planners and decision makers to assess the quality of their integrated urban sustainability and climate change plans. It is worth noting that there was no significant difference in scoring between urban planning and urban geography scholars, perhaps because the scoring criteria were more intuitive; as mentioned earlier, scholars assigned scores to indicators from zero to two based on the presence or absence of mentions, detailed discussions, and descriptions of the indicator. There are only a few criteria where scholars differed slightly in their scoring, for example the percentage of energy from fossil fuels (natural gas, gasoline etc.)/hydro or other water-intensive energy sources, the percentage of hardscape with a surface reflectance index (SRI) >30, whether or not the plan addresses climate change adaptation, whether or not the plan’s approach to flooding considers accommodation, whether the integration of adaptation and mitigation is acknowledged, and other criteria. Urban geographers tended to give two points for specific data, while some urban planning scholars were more concerned with the level of detail in the engineering categories of strategies to judge their scores. Overall, no significant differences were found between scholars with Ph.D.s and those with masters in the scoring for this assessment framework. This also indicates the general applicability and operability of the assessment framework.

We attempted to make recommendations for aspects of the Arden Structure Planning Assessment that are underperforming. In 2021, the census resident population of Arden was 413, living in 224 dwellings with an average household size of 2.11 [58]. Melbourne’s population is estimated to grow to 30 million by 2100 [58]. Arden has an abundance of land, and the new railway station will provide opportunities for local development and population growth. The plan takes into account the provision of 6% affordable housing for low- and middle-income households, which will also avoid gentrification caused by population growth and contribute to the sustainability of the city’s population. The Victorian government is aware of the impact of climate change on urban sustainability and has an ambition to address it. However, the presentation of the context and impacts of urban sustainability and climate change should be more regionally specific and provide a clearer explanation of the concepts, which will help non-specialists to understand the significance of and need for relevant planning. The future development of Arden as a regional centre with a new railway station and a large number of buildings and roads with high heat-storage capacities will inevitably result in the area becoming a new urban heat island in North Melbourne. The plan should recognise this before it happens and propose specific approaches to avoid the formation of a new urban heat island, including strategies for the construction of physical spaces such as buildings and roads, for example, that use the absorption and reflection of solar energy by colour to reduce their energy consumption. It should also propose approaches for promoting the use of a variety of public transportation options. In addition, it is also important to promote science and technology and to guide people toward developing ecological and energy-saving habits in their lives [59]. Different environmental behaviours can be encouraged or punished through a clear system of rewards and penalties. For example, Singapore’s comprehensive eco-accountability system [60] relies on effective government policies, public participation, and transparent information to create a beautiful natural environment and a gardenlike city [61].

The lack of analysis in the Arden Structure Plan provides little guidance for the preparation and practice of planning. The plan scored lowest on the analysis factor and lacked specific planning strategies and measures related to urban sustainability and the response to climate change. According to the three ethical imperatives for sustainable urban development [34], the environmental resources of the city are limited in terms of environmental limits; thus, the development model of the city will be increasingly fine-tuned and trend towards information-based management. The Arden plan lacks a large amount of basic data on the urban environment to support it. Big data technology can be used to collect, collate, and analyse data and information about the urban environment to provide a more effective scientific strategy for the prevention of urban natural disasters. Only with a full understanding of the natural environmental base of the area can the boundaries of development be clarified to avoid encroaching on the environmental resources that should be left to future generations. Regarding human needs, the Arden Structure Plan (2022) is inadequate in terms of crime-prevention measures; however, urban safety is one of the most fundamental human needs. Cozens proposed that ‘designing out crime’ represents an important tool to help develop urban sustainability [62]. A city cannot be called sustainable if its residents are concerned about safety [63]. The Arden plan, which only proposes to promote street surveillance with a layer of buildings around the main pavements, is incomplete. Therefore, the government should consider developing products and systems that are more resistant to crime to reduce its impact and should also use the built environment to reduce waste generation, promote liveability, improve the quality of life, and reduce opportunities for crime. In terms of social equity, the analysis showed that participation in the plan by residents and whether the plan embraced and recognised differences both scored zero. Berke et al. stated that a quality plan should include a stakeholder-engagement approach [24]. In addition, according to Tan and Artist [64], citizen participation is necessary because public services are not the same as private sector services. The Arden Structure Plan (2022) clearly lacks a specific approach to enhancing community participation and does not focus on how special groups and vulnerable populations can actually be involved in the development of plans. Sustainable cities must respect human rights and be planned and designed with people in mind. The community should strengthen its connection with residents, publicise and share relevant policy information in a timely manner, and enhance public opinion surveys to ensure that decisions can truly serve and benefit all residents. This will also help to foster a sense of responsibility and territorial belonging for the development of Arden, such that residents will feel that everyone has a voice in decision making and that the community operates in an equitable manner.

In 2021, UNEP stated that global temperatures will rise by 2.7 °C by 2100, which is well above the Paris Agreement’s temperature control target. The Arden Structure Plan (2022) proposes net-zero emissions across the precinct by 2040, and all planning decisions and active transport options will support this goal. The government seems to be ambitious and prepared to adapt to mitigate the impacts of climate change in Arden. The planned mixed-land-use pattern will effectively reduce the environmental damage caused by urban construction, and active transport options—mainly walking and cycling—will reduce environmental pollution. All these measures will make Arden a suitable place for population growth in the context of climate change. Measures to address climate change mainly include mitigation, adaptation, and the integration of the two. Climate change mitigation targets should contain both qualitative and quantitative characteristics, such as specific warming limits and data on greenhouse gas emission reductions, which are lacking in the Arden Structure Plan (2022). According to Albrechts, there is a risk of losing momentum if there are no short-term goals to be achieved [65]. The plan should therefore establish a schedule of specific mitigation targets for different periods to ensure that its vision can be achieved over time. In addition, less attention is given to adaptation than to mitigation in planning. However, the planning should take full account of the particularities of the Arden environment, which is vulnerable to disturbances from multiple sources of flooding, and should propose specific measures to adapt to this regional specificity. The plan presents a good vision of water in Arden as a landscape feature through innovative flood-management solutions. However, the basic data and measures of the flood-protection plan are not detailed enough, and the management of flood-protection stocks, the emergency management mechanisms, the construction of flood-protection works, and the delineation of safety zones need to be presented more clearly. Addressing climate change is not the sole responsibility of one sector; mitigation and adaptation measures need to work together, and all relevant sectors should share the impacts of climate change. The 3S technique was used to analyse floodplains and the plan’s water-ecological zoning to draw conclusions about its water-ecological sensitivity in terms of setting up soil and water conservation forests between built-up areas and rivers and lakes to achieve water retention and filtration and to guide site layout [66]. Flooding can have direct and indirect impacts on transport facilities in Arden, and heavy rainfall also affects the way people move around. In the face of emergency climate change situations, the transport department and the meteorological department should liaise in a timely manner to take emergency warning measures and inform residents in advance of appropriate travel options to mitigate the effects of special climate change in a timely manner. At the same time, the routine maintenance of transport facilities should be carried out to adapt their materials and permeability to climate change. For example, Stockholm deals with heavy rainfall by using green roofs in buildings, which are porous surfaces for on-site infiltration that treat rainwater for new water-storage technologies and green spaces, rather than discharging it into the sewage system [67]. In addition, integrating green and grey infrastructure (traditional municipal facilities) into one sustainable stormwater management system can address land-use constraints, fulfil the stormwater function of green infrastructure [68], enhance urban-plaza landscaping, and increase the public accessibility of water. This will help Arden mitigate and adapt to the impacts of extreme weather on the urban environment and on people’s lives as well as reduce property damage.

The most important area for improvement in the action section is the low-scoring monitoring and evaluation indicator. Climate change is an unpredictable process, and planning is dynamic in nature. Big data is not only used for climate change observation, monitoring, early warning, and trend prediction, but also for urban disaster identification and assessment and public engagement [69,70]. Big data can provide important support for the science of planning, decision making, implementation, and evaluation, both in terms of research paradigms and technical methods, and has great potential for the development of climate change-adaptation planning strategies [71,72]. The Arden Structure Plan (2022) needs to strengthen the way it is monitored and evaluated in order to obtain real-time data and adjust its content in a timely manner. The continuous tracking of implemented activities will also help to determine the extent to which targets are being met [73].

## 5. Conclusions

The Victorian government is confident in the development of Arden as a city and plans to provide more affordable housing and employment opportunities in anticipation of its population growth. However, there are still issues with the plan, such as a lack of data and time-bound development targets, unclear public engagement, low attention to urban crime, and insufficient adaptation measures to address climate change. With the uncertainty and complexity surrounding urban climate change, strengthening urban resilience through mitigation and adaptation measures to address the impacts of climate change on metropolises and their people in order to achieve sustainable cities will become a necessary journey for future urban development. Here, we assessed the content and quality of the Arden Structure Plan (2022) in terms of urban sustainability and climate change and evaluated the suitability of Arden as a precinct for population growth in the context of climate change, which will contribute to the adaptation and refinement of Arden’s future development plans.

The results of our study showed that there were significant differences in the evaluation scores of Arden’s structural plan in the three areas of awareness, analysis, and action. The action factor scored the highest, indicating that action-related concepts were more fully described in the plan. The analysis factor scored the lowest, indicating that analysis was not sufficiently featured in the plan. This suggests that the Arden Structure Plan (2022) has clear steps for implementation, but its analysis needs to be more specific and systematic. Based on this, we presented targeted recommendations for improving the Arden Structure Plan, especially in terms of analysis, with suggestions for the analysis of sustainable urban development based on environmental constraints, human needs, and social equity, as well as specific recommendations related to the analysis of climate change.

This study provides an assessment framework and methodology for the evaluation of urban sustainability and climate change planning, providing guidance for the assessment of water-sensitive urban planning for urban regeneration development. It contributes to research on planning policies for sustainable development and climate change in water-sensitive cities. We recommend that planners use urban sustainability and climate change ex ante evaluation as a key step in the comparison of planning options; it will act as an important guide to whether or not a plan is geographically relevant and scientifically sound. However, there are still some limitations to this study. Firstly, although we tried to be as objective as possible, the different levels of contextual knowledge of the foreign experts still did not prevent the scoring protocols from being somewhat biased and subjective, which had an impact on the assessment results. Secondly, the fact that we did not assess both the Melbourne and Victorian plans did not allow us to evaluate the longitudinal consistency of the urban sustainability and climate change indicators; however, this had only a very minor impact on the assessment of the Arden Structure Plan. Finally, the results of the evaluation of the Arden Structure Plan were discussed and relevant recommendations were made; however, the recommendations were mostly qualitative and less quantitative. Therefore, there are still some shortcomings of the Arden Structure Plan that must be improved, which is what motivates us to continue this work.

## Figures and Tables

**Figure 1 ijerph-20-02469-f001:**
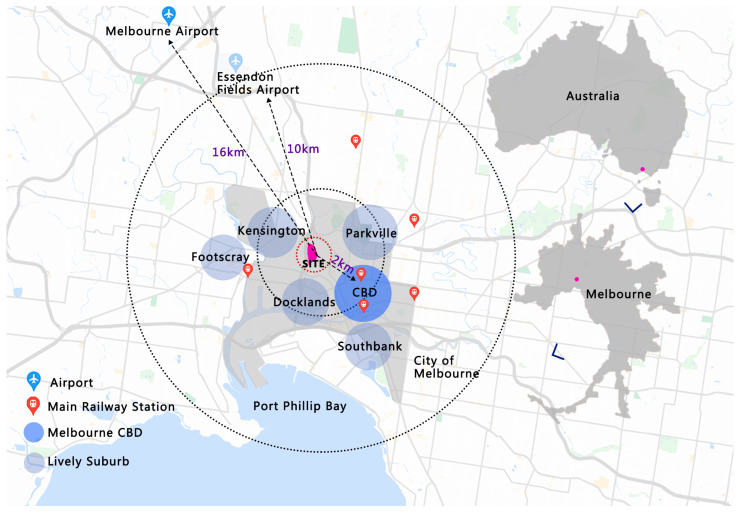
Arden Location Map.

**Figure 2 ijerph-20-02469-f002:**
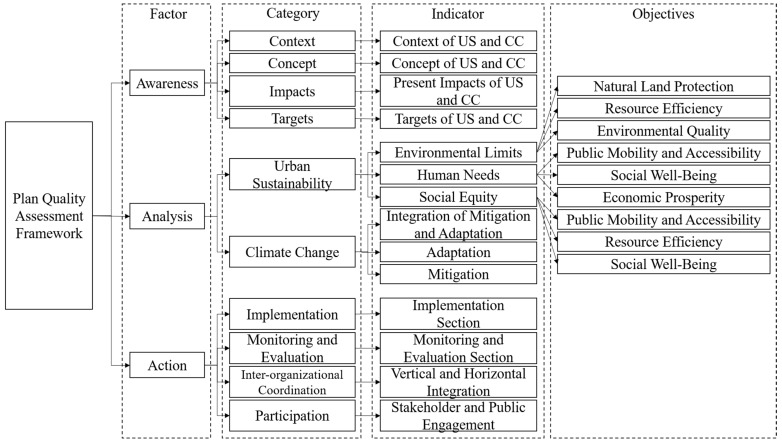
Plan quality assessment framework.

**Figure 3 ijerph-20-02469-f003:**
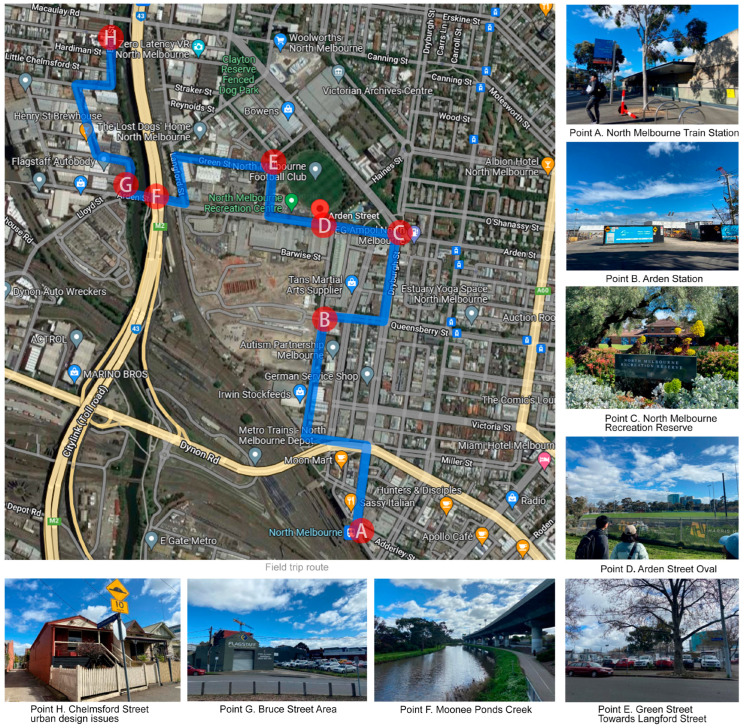
Fieldwork route and photos.

**Figure 4 ijerph-20-02469-f004:**
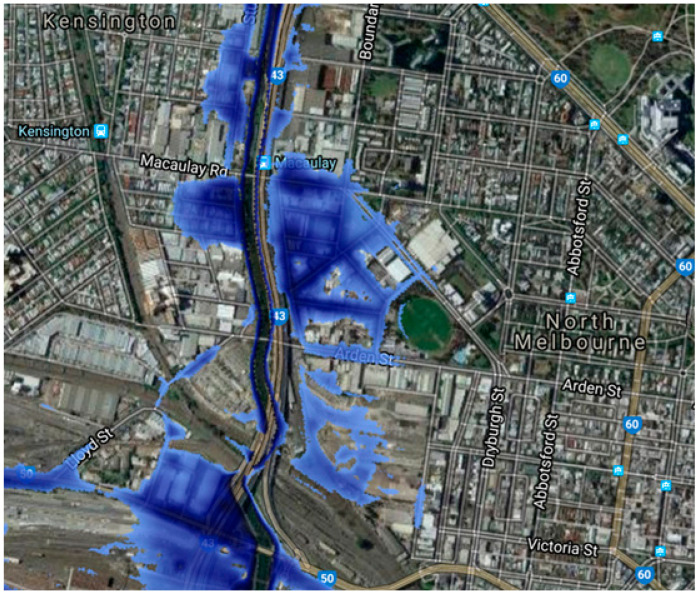
Arden’s flood area with an increase in sea level of 2.7 m. Source: foreground, 2018, https://www.foreground.com.au/planning-policy/reimagining-australias-temperate-kakadu/ (accessed on 20 September 2022).

**Figure 5 ijerph-20-02469-f005:**
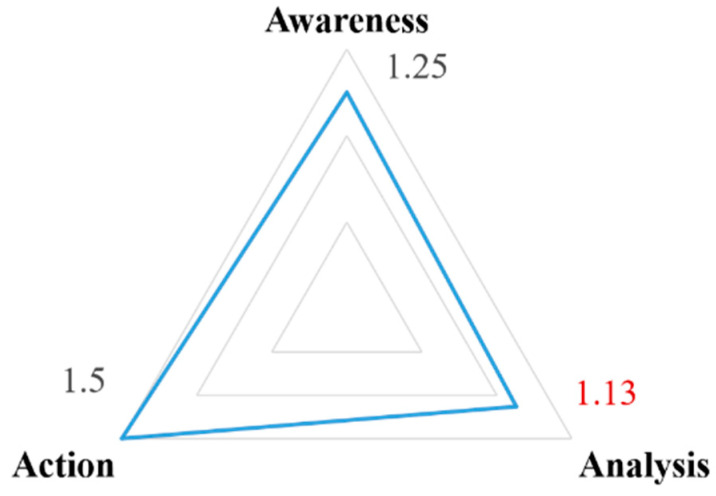
Analysis of core factors.

**Figure 6 ijerph-20-02469-f006:**
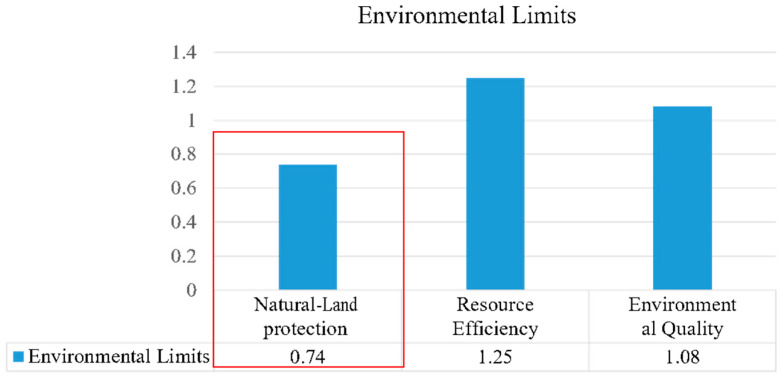
Analysis of environmental limits.

**Figure 7 ijerph-20-02469-f007:**
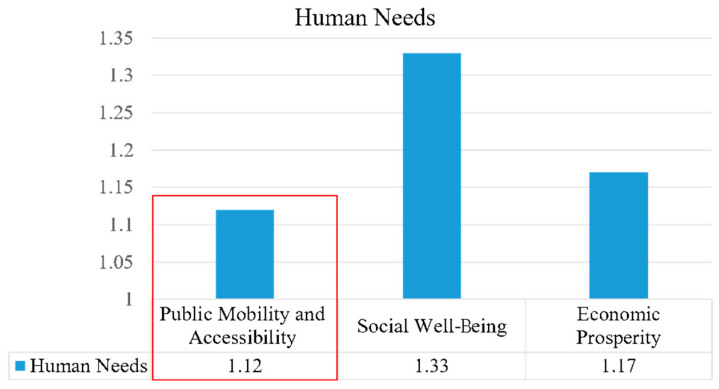
Analysis of human needs.

**Figure 8 ijerph-20-02469-f008:**
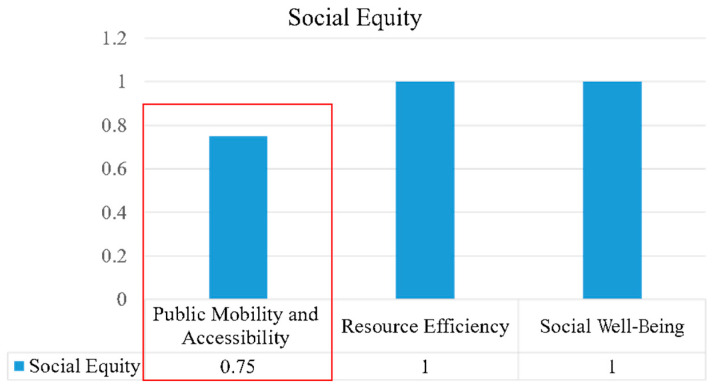
Analysis of social equity.

**Figure 9 ijerph-20-02469-f009:**
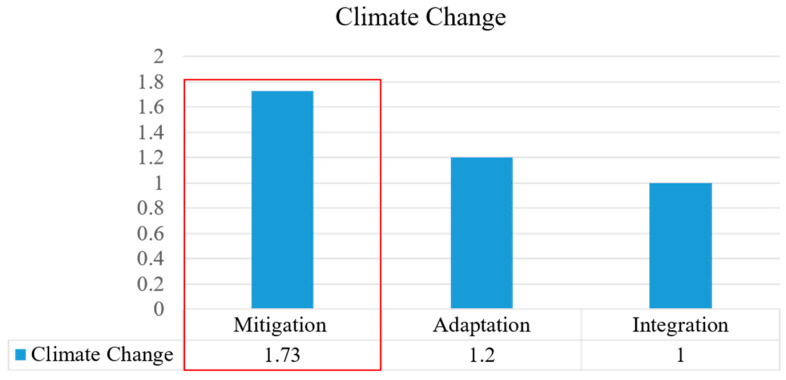
Analysis of climate change evaluation.

## Data Availability

The data presented in this study are available on request from the corresponding author.

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
