# Peer review of "The Effectiveness of Local Governments’ Policies in Response to Climate Change: An Evaluation of Structure Planning in Arden, Melbourne"

_ijerph, 2023, doi:10.3390/ijerph20032469_

Round 1
Reviewer 1 Report
Article has serious flaws. The cited references are few and not relevant to the research. The methodology used is limited. The results lack of novelty. No discussion neither comparison with other studies is given. Conclusions are very local.
Please find the specific comments in the attachment.

Author Response
Dear Editor Zhou and reviewers,
Thank you for your letter and the reviewers’ comments on our manuscript entitled " The Effectiveness of Local Governments’ Policies in Response to Climate Change: An Evaluation of Structure Planning in Arden, Melbourne " (ID: ijerph-2123020). Those comments are very helpful for revising and improving our paper, as well as the important guiding significance to another research. We have studied the comments carefully and made corrections which we hope meet with approval. The main corrections are in the manuscript and the responds to the reviewers’ comments are as follows (the replies are highlighted in blue).
Reviewer 1:
The main objective of the paper is to evaluate the Arden Structure Plan critically and make suggestions on how to make the plan's urban sustainability and climate change considerations better. However, this evaluation is very subjective. The evaluation is based on scoring three indicators (factors), awareness, analysis, and action, according to how these indicators were described or evidenced throughout the plan. So, the scoring given by the authors is subjective since depends on how the authors consider these indicators were mentioned in the Plan. In the results and discussion section, the authors only describe the score for each indicator. The values reported for each score were not obtained rigorously and no comparison or discussion is provided.
The topic is original, but this manuscript does not accomplish the scientific principles and research practices (such as formulating a hypothesis, designing an experiment to test the hypothesis, and collecting and interpreting data).
As it was mentioned before, the results obtained in this manuscript were not compared with other published material. The cited references in the introduction, methodology, and results sections are few and not relevant to the research.
Several specific improvements can be provided to the authors. The most important improvements are:
- Figure 1. Several methodologies are given in the literature for flood risk mapping. The authors just depicted a flood risk map without an explanation of how this map was obtained.
- Figure 2. This figure was generated after fieldwork. This map should be part of the results of the manuscript. The authors should explain why this map is important to obtain the indicators’ scoring.
- Several multi-criteria methodologies are provided in the literature to assess urban sustainability and climate change for decision-makers. Why did the authors choose the methodology proposed in this study? A discussion should be provided by the authors.
- No. In lines 247 – 249, the authors suggest that the results obtained in this paper include the consideration of enhancing human capabilities, eradicating extreme poverty, ensuring ‘rich’ participation, ensuring fair distribution, mitigating climate change, and safeguarding biosphere integrity. However, this situation is not evidenced in the Manuscript.
- The conclusions are very local. I consider that this paper should highlight how the methodology proposed in this manuscript can be used for the assessment of other plans worldwide.
- As it was mentioned before, the cited references in the introduction, methodology, and results sections are few and not relevant to the research. Besides, a scoring comparison of the results with other published material should be provided.
- Figures 4, 5, 7, and 9 are not relevant. Consider removing them.
Response:
Thank you very much for your comments. We have revised the manuscript and invited native English speakers to edit the language, and we proofread the entire text. The language editing certificate has also been submitted.
- In response to your comments, we have explained the source of this figure. The details are as follows.
“According to Figure 4, in the region’s draught vision framework, a sizable portion of land will be at risk of flooding, which will undoubtedly become more severe if sea levels rise by 2.7 m, as described in the worst-case scenario [51]. Point G and point H are outside the reach of the Arden Structure Plan; however, it is worth noting that the residential houses in these areas are also vulnerable to flooding.
Figure 4. Arden’s flood area with an increase in sea level of 2.7 metres. Source: foreground, 2018, https://www.foreground.com.au/planning-policy/reimagining-australias-temperate-kakadu/ (accessed on 20 September 2022).”
- In response to your comments, we have rewritten the scoring protocol section to explain the process in more detail. The details are as follows.
“2.2.2. Fieldwork and Scoring Protocol
This study invited 15 Melbourne- and Beijing-based urban planning and urban geography scholars, 10 of which held PhDs, including one professor, three associate professors, six assistant professors, and four current master’s students. In order to ensure an objective understanding and real perception of Arden’s current situation, and to increase the accuracy and reliability of the scoring results, four members of the study carried out fieldwork on 16 September 2022; the route can be seen in Figure 3. Point A is the North Melbourne train station, which is currently the closest train station to Arden. The streets here will provide quick and convenient walking and cycling connections between key spaces such as the Macaulay, Arden, and North Melbourne train stations, as well as open spaces within and outside of Arden. Point B is the new Arden station, due to open in 2025, which will accelerate Arden’s emergence as the centre of the district and assist in achieving the target of at least 60% public transport, 30% walking and cycling, and no more than 10% private vehicles. Points C, D, and E are the North Melbourne Recreation Reserve, underground storage tanks, and temporary parking, respectively. Flows from Melbourne’s inner northwest have an impact on the Moonee Ponds Creek (point F). The current drain-age system is not designed to handle this amount of stormwater, resulting in dangerous flood depths. According to Figure 4, in the region’s draught vision framework, a sizable portion of land will be at risk of flooding, which will undoubtedly become more severe if sea levels rise by 2.7 m, as described in the worst-case scenario [51]. Point G and point H are outside the reach of the Arden Structure Plan; however, it is worth noting that the residential houses in these areas are also vulnerable to flooding.”
- In response to your comments, we have rewritten the introduction and methods sections. In the introduction we have added a literature review that describes the types of assessment methods. In the methods section we summarize the relevant assessment methods. Given that this assessment of the Arden Structure Plan is a combination of both sustainability and climate change, we have drawn on the AAA assessment framework of Elgendawy et al. to develop an assessment framework that incorporates the three ethical requirements of sustainable development. The modifications are as follows.
“In the last hundred years, the world has experienced significant changes, mainly characterised by warming, with widespread and far-reaching effects [4-6]. In Australia, particularly in metropolitan areas such as Sydney and Melbourne, the pressures of population growth and the effects of climate change compound the problems of environmental degradation, high resource consumption, and the loss of development space [7,8]. The United Nations Environment Programme (2021) claims that as emissions pledges be-tween nations increase in size and complexity, climate change becomes a more serious issue [9]. Additionally, national governments are unable to make a significant contribution towards effectively reducing greenhouse gas emissions (for example, by repealing Australia’s carbon tax). As a result, local governments play a crucial role in directing sustainable development. As of 2022, the population of Melbourne is about 5 million, and the population density in the city is about 500 residents per square kilometre. In fact, around 75% of the population of Victoria lives in Melbourne. Melbourne is not only a populated city, but it is also expanding quickly. Melbourne is expected to overtake Sydney to become Australia’s largest city in about 25 years if it continues to grow at this rate [10]. The structure plan must take on additional functions not only to ensure the economic and social vitality of the area, but also to develop a range of tools to help local governments create and manage their plans to make them more resilient to climate change and increase the liveability of these regions [11].
Given that planning decisions have a decades-long impact on both the natural and built environments, which is sometimes irreversible [12], conducting a programme assessment is a critical step in ensuring that these plans produce the desired results [13,14]. Planning assessments can help to increase the accountability of local governments and public institutions and inform decision makers on ways to address policy issues [15], ensuring that the planning is serious and scientific [16]; in addition, they also strengthen public confidence in planning decisions [17,18]. Scholars have developed analytical methods and conducted empirical studies to assess the quality of planning [19,20]. The question of whether planning assessments should focus on ‘ex ante (or a priori)’, ‘on going’, or ‘and ex post’ has been widely discussed by scholars in various countries [21-25]. Although quantitative indicators play a pivotal role, the joint assessment of qualitative description means is comprehensive, especially for those planning implementation goals that cannot be directly quantified, where qualitative indicator description is also essential [26]. In the 1980s, Canadian scholar Hok-Lin Leung proposed the S-CAD (subjectivity, consistency, adequacy, and dependency) public policy evaluation method, which helps policy-making participants develop and analyse their own policy recommendations as well as evaluate and resolve their differences with other participants ([27-29]. Many governments have also established planning evaluation systems and applied planning evaluation methods in practical work. The UK represents the Western country with the most complete planning evaluation system, and this system was eventually adopted by the EU, forming a regional planning policy evaluation system under the EU framework [30]. Port-land’s Comprehensive Plan Assessment Plan, based on an analysis of the implementation environment and urban development trends of Portland’s 1980 master plan, focuses on assessing the timeliness of the implementation of the plan and provides a systematic assessment of the master plan in seven areas: economic development, environmental protection, housing planning, infrastructure planning, sustainable development, transportation, and urban form [31]. Hong Kong’s Territorial Development Strategy Review focuses on assessing the forward-looking nature of planning, particularly in terms of assessing Hong Kong’s competitiveness in the global urban system [32,33].”
“2.2. Method
2.2.1. Assessment Framework
As previously mentioned, there has been a growing interest among planning scholars in evaluating the quality of a range of plans, especially those focusing on natural hazards [45], ecosystem management [46], climate change [47], and sustainable development [48]. However, there is a lack of common criteria and frameworks for assessing the quality of these plans. Kaiser et al. highlighted three core characteristics that constitute a high-quality framework, including a strong factual basis, clear goals, and appropriate policies [49]. These characteristics can be used to assess the quality of a plan in view of a strategic direction and statement via a quantitative method [50]. Berke et al. expanded these three characteristics to eight characteristics, including fact base, goals, policies, implementation, monitoring and evaluation, inter-organizational coordination, participation, and plan organization and presentation [24]. Elgendawy et al. adopted an AAA approach based on three critical components including awareness (targets and analysis), basis, and action (implementation and monitoring to assess plan quality) [50].
In order to assess the appropriateness of the sustainability and climate change considerations outlined in Arden’s plan and the feasibility of its actions, this study conducted a plan quality evaluation by assessing the content of the plan with respect to sustainability and climate change. Here, the AAA approach was adapted to form a new plan quality assessment protocol focusing on these two themes. The overall framework is illustrated in Figure 2. It covers three core factors including awareness, analysis, and action. For the ‘Awareness’ factor, there are four indicators including context, concept, impacts, and tar-gets. For the ‘Analysis’ factor, we adapted an NSA framework [37] and a climate change assessment framework [41] to measure the consideration of these two themes and specific quantitative indicators in the plan. Although this NSA framework is focused on assessing the current state of sustainability within a site, the criteria were used as a reference in this study to measure whether they were considered and developed in the Arden Structure Plan. This NSA has six objectives: resource efficiency, economic prosperity, social well-being, public mobility and accessibility, environmental quality, and natural land protection. These six objectives are based on three moral imperatives, satisfying human needs, ensuring social equity, and respecting environmental limits, and they are adapted from the six themes identified by Holden et al. [34]. In addition, referring to Hurlimann et al. [41], the climate change assessment framework used here consists of three parts: mitigation, adaptation, and integration. It focuses on actions to address climate change in the Arden Structure Plan and highlights the considerations at the state level and on a smaller scale. The actions component of this framework identifies appropriate measures and monitoring strategies to demonstrate how the local government will achieve sustainable urban development and reduce the trend of climate change, which include implementation, monitoring, and evaluation as well as inter-organizational coordination and participation. In this study, we combined both qualitative and quantitative methods to develop the plan quality assessment framework. The content of the Arden Structure Plan was analysed by a description evaluation and scored according to the scoring protocol. Here, we present an overall view and conclusion of the plan’s quality, focusing on the outcomes of the urban sustainability assessment and climate change assessment, and discuss the shortcomings and potential of the proposed development in Arden.”
- Thank you for the reminder. Here is our mistake, our language caused ambiguity in the cited literature. We have made the correction. The modifications are as follows.
“In addition, referring to Hurlimann et al. [41], the climate change assessment framework used here consists of three parts: mitigation, adaptation, and integration.”
- In response to your comments, we have rewritten the concluding section in order to emphasize the significance and application of the methods presented in this manuscript. The modifications are as follows.
“5. Conclusions
The Victorian Government is confident in the development of Arden as a city and plans to provide more affordable housing and employment opportunities in anticipation of population growth. However, there are still issues with the plan, such as a lack of data and time-bound development targets, unclear public engagement, low attention to urban crime, and insufficient adaptation measures to address climate change. With the uncertainty and complexity surrounding urban climate change, strengthening urban resilience through mitigation and adaptation measures to address the impacts of climate change on metropolises and their people in order to achieve sustainable cities will become a necessary journey for future urban development. Here, we assessed the content and quality of the Arden Structure Plan (2022) in terms of urban sustainability and climate change and evaluated the suitability of Arden as a precinct for population growth in the context of climate change, which will contribute to the adaptation and refinement of Arden’s future development plans.
The results of our study showed that there were significant differences in the evaluation scores of Arden’s structural plan in the three areas of awareness, analysis, and action. The action factor scored the highest, indicating that action-related concepts were more fully described in Arden’s structure plan. The analysis factor scored the lowest, indicating that analysis was not sufficiently featured in the plan. This suggests that the Arden Structure Plan (2022) has clear steps for implementation, but its analysis needs to be more specific and systematic. Based on this, we presented targeted recommendations for improving the Arden Structure Plan, especially in terms of analysis, with suggestions for the analysis of sustainable urban development based on environmental constraints, human needs, and social equity, as well as specific recommendations related to the analysis of climate change.
This study provides an assessment framework and methodology for the evaluation of urban sustainability and climate change planning, providing guidance for the assessment of water-sensitive urban planning for urban regeneration development. It contributes to research on planning policies for sustainable development and climate change in water-sensitive cities. We recommend that planners use urban sustainability and climate change ex ante evaluation as a key step in the comparison of planning options; it will act as an important guide to whether or not a plan is geographically relevant and scientifically sound. However, there are still some limitations to this study. Firstly, although we tried to be as objective as possible, the different levels of contextual knowledge of the foreign experts still did not prevent the scoring protocols from being somewhat biased and subjective, which had an impact on the assessment results. Secondly, the fact that we did not assess both the Melbourne and Victorian plans did not allow us to evaluate the longitudinal consistency of the urban sustainability and climate change indicators; however, this had only a very minor impact on the assessment of the Arden Structure Plan. Finally, the results of the evaluation of the Arden Structure Plan were discussed and relevant recommendations were made; however, the recommendations were mostly qualitative and less quantitative. Therefore, there are still some shortcomings of the Arden structure plan that must be improved, which is what motivates us to continue this work.”
- In response to your comments, we have modified the introduction, methods, and results chapters, added references, and emphasized a bit of consistency of ideas and conclusions with other studies. The modifications are as follows.
“1. Introduction
Structure planning is a type of spatial planning and is a part of urban planning practices in the United Kingdom and Australia [1]. In Australia, structure plans are commonly prepared at the subregional, district, and local levels under strategic planning. Structure plans set the direction of development in urban sub-regions and play an important role in the implementation and effectiveness of strategic policy objectives. They also make provisions for changing community needs and set out how they should be managed by local government or councils. They guide major changes in land use, transportation integration, built form, public space, and cultural transmission, working together to achieve the eco-nomic, social, and environmental objectives of the area [2,3].
In the last hundred years, the world has experienced significant changes, mainly characterised by warming, with widespread and far-reaching effects [4-6]. In Australia, particularly in metropolitan areas such as Sydney and Melbourne, the pressures of population growth and the effects of climate change compound the problems of environmental degradation, high resource consumption, and the loss of development space [7,8]. The United Nations Environment Programme (2021) claims that as emissions pledges be-tween nations increase in size and complexity, climate change becomes a more serious issue [9]. Additionally, national governments are unable to make a significant contribution towards effectively reducing greenhouse gas emissions (for example, by repealing Australia’s carbon tax). As a result, local governments play a crucial role in directing sustainable development. As of 2022, the population of Melbourne is about 5 million, and the population density in the city is about 500 residents per square kilometre. In fact, around 75% of the population of Victoria lives in Melbourne. Melbourne is not only a populated city, but it is also expanding quickly. Melbourne is expected to overtake Sydney to become Australia’s largest city in about 25 years if it continues to grow at this rate [10]. The structure plan must take on additional functions not only to ensure the economic and social vitality of the area, but also to develop a range of tools to help local governments create and manage their plans to make them more resilient to climate change and increase the liveability of these regions [11].
Given that planning decisions have a decades-long impact on both the natural and built environments, which is sometimes irreversible [12], conducting a programme assessment is a critical step in ensuring that these plans produce the desired results [13,14]. Planning assessments can help to increase the accountability of local governments and public institutions and inform decision makers on ways to address policy issues [15], ensuring that the planning is serious and scientific [16]; in addition, they also strengthen public confidence in planning decisions [17,18]. Scholars have developed analytical methods and conducted empirical studies to assess the quality of planning [19,20]. The question of whether planning assessments should focus on ‘ex ante (or a priori)’, ‘on going’, or ‘and ex post’ has been widely discussed by scholars in various countries [21-25]. Although quantitative indicators play a pivotal role, the joint assessment of qualitative description means is comprehensive, especially for those planning implementation goals that cannot be directly quantified, where qualitative indicator description is also essential [26]. In the 1980s, Canadian scholar Hok-Lin Leung proposed the S-CAD (subjectivity, consistency, adequacy, and dependency) public policy evaluation method, which helps policy-making participants develop and analyse their own policy recommendations as well as evaluate and resolve their differences with other participants ([27-29]. Many governments have also established planning evaluation systems and applied planning evaluation methods in practical work. The UK represents the Western country with the most complete planning evaluation system, and this system was eventually adopted by the EU, forming a regional planning policy evaluation system under the EU framework [30]. Port-land’s Comprehensive Plan Assessment Plan, based on an analysis of the implementation environment and urban development trends of Portland’s 1980 master plan, focuses on assessing the timeliness of the implementation of the plan and provides a systematic assessment of the master plan in seven areas: economic development, environmental protection, housing planning, infrastructure planning, sustainable development, transportation, and urban form [31]. Hong Kong’s Territorial Development Strategy Review focuses on assessing the forward-looking nature of planning, particularly in terms of assessing Hong Kong’s competitiveness in the global urban system [32,33].
Sustainable development and combating climate change are currently some of the most topical issues in urban research and are among the core objectives of urban development. Planning assessment methodologies that address sustainable urban development, climate change, and the maintenance of infrastructure support and ecological and green spaces have been developed, for example, the Urban Sustainability Assessment Framework and the Climate Change Assessment Framework. Holden et al. suggested that urban sustainability should be based on three imperatives: satisfying human needs, ensuring social equity, and respecting environmental limits, identified by six key themes [34]. This emphasizes the primary objective and proposes quantifiable indicators, which makes this definition more forward-looking and feasible both at the state level and municipality levels. Although the neighbourhood is considered the most appropriate spatial scale for improving sustainable built environments [35], bringing sustainability to small neighbourhoods has been acknowledged as a challenge for developers and local municipalities due to a lack of institutional and financial capacities. Compared with larger communities, smaller neighbourhoods are limited in their ability to adopt sustainable practices. Neighbourhood sustainability assessment (NSA) tools have been introduced to measure the effectiveness of sustainability development at this spatial scale [36], including LEED (Leadership in Energy and Environmental Design), BREEAM (Building Re-search Establishment Environmental Assessment Method), and ITACA (Italian Institute for Innovation and Transparency in Procurement and Environmental Compatibility). Haider et al. developed a sustainability assessment framework for a small urban neighbourhood based on the analysis of several NSA tools [37]. These NSA tools are considered reliable in assessing the potential for sustainable development in precincts and achieving sustainability goals [38]. According to this feature, sustainability assessment tools can al-so be used to assess the degree of consideration of sustainable development in specific strategic plan documents. The consideration of climate change in municipalities’ plans provides a crucial opportunity to address this global issue, as cities are recognized as major drivers in addressing climate change actions. Larsen and Kørnøv presented three over-all approaches to address climate change: mitigation, adaptation, and baseline adaptation [39]. Grafakos et al. indicated the significance of integrating mitigation and adaptation for a global transition to respond to climate change [40]. This can help avoid conflict between different policies and bring far-reaching benefits to enhance collaboration. Mitigation is considered as a global-scale issue. However, with further research and the advancement of strategic climate change plans, there is a growing awareness of the impact of bottom-up behaviour on mitigation action. Thus, it is important to consider mitigation at both the global level and local municipal levels. In addition, considering the role of adaptation and integration is also crucial in creating climate change assessment frameworks. In order to assess the effectiveness and advancement of these plans in addressing climate change, Hurlimann et al. introduced an assessment focusing on the actions of mitigation, adaptation, and integration [41].
The quality of the plan is a key determinant of its outcome, and a high-quality plan is more likely to produce positive outcomes [42]. The five million residents in Melbourne deal with serious environmental issues, including drought, bushfires, floods, and heat-waves [43] , especially in Arden, which experiences severe flooding. Therefore, it is necessary to critically evaluate the structure plan for the Melbourne urban renewal area of Arden. The government is looking for suggestions on how to better take climate change and urban sustainability into account in the plan and whether the urban renewal project should move forwards. The primary objective of this study was to evaluate the Arden Structure Plan and provide suggestions on how to improve the plan’s urban sustainability and climate change considerations. Secondly, we aimed to offer guidance on whether Arden is a suitable location for Melbourne’s population growth considering the impacts of climate change anticipated to occur by 2100.”
“2.2. Method
2.2.1. Assessment Framework
As previously mentioned, there has been a growing interest among planning scholars in evaluating the quality of a range of plans, especially those focusing on natural hazards [45], ecosystem management [46], climate change [47], and sustainable development [48]. However, there is a lack of common criteria and frameworks for assessing the quality of these plans. Kaiser et al. highlighted three core characteristics that constitute a high-quality framework, including a strong factual basis, clear goals, and appropriate policies [49]. These characteristics can be used to assess the quality of a plan in view of a strategic direction and statement via a quantitative method [50]. Berke et al. expanded these three characteristics to eight characteristics, including fact base, goals, policies, implementation, monitoring and evaluation, inter-organizational coordination, participation, and plan organization and presentation [24]. Elgendawy et al. adopted an AAA approach based on three critical components including awareness (targets and analysis), basis, and action (implementation and monitoring to assess plan quality) [50].
In order to assess the appropriateness of the sustainability and climate change considerations outlined in Arden’s plan and the feasibility of its actions, this study conducted a plan quality evaluation by assessing the content of the plan with respect to sustainability and climate change. Here, the AAA approach was adapted to form a new plan quality assessment protocol focusing on these two themes. The overall framework is illustrated in Figure 2. It covers three core factors including awareness, analysis, and action. For the ‘Awareness’ factor, there are four indicators including context, concept, impacts, and tar-gets. For the ‘Analysis’ factor, we adapted an NSA framework [37] and a climate change assessment framework [41] to measure the consideration of these two themes and specific quantitative indicators in the plan. Although this NSA framework is focused on assessing the current state of sustainability within a site, the criteria were used as a reference in this study to measure whether they were considered and developed in the Arden Structure Plan. This NSA has six objectives: resource efficiency, economic prosperity, social well-being, public mobility and accessibility, environmental quality, and natural land protection. These six objectives are based on three moral imperatives, satisfying human needs, ensuring social equity, and respecting environmental limits, and they are adapted from the six themes identified by Holden et al. [34]. In addition, referring to Hurlimann et al. [41], the climate change assessment framework used here consists of three parts: mitigation, adaptation, and integration. It focuses on actions to address climate change in the Arden Structure Plan and highlights the considerations at the state level and on a smaller scale. The actions component of this framework identifies appropriate measures and monitoring strategies to demonstrate how the local government will achieve sustainable urban development and reduce the trend of climate change, which include implementation, monitoring, and evaluation as well as inter-organizational coordination and participation. In this study, we combined both qualitative and quantitative methods to develop the plan quality assessment framework. The content of the Arden Structure Plan was analysed by a description evaluation and scored according to the scoring protocol. Here, we present an overall view and conclusion of the plan’s quality, focusing on the outcomes of the urban sustainability assessment and climate change assessment, and discuss the shortcomings and potential of the proposed development in Arden.
2.2.2. Fieldwork and Scoring Protocol
This study invited 15 Melbourne- and Beijing-based urban planning and urban geography scholars, 10 of which held PhDs, including one professor, three associate professors, six assistant professors, and four current master’s students. In order to ensure an objective understanding and real perception of Arden’s current situation, and to increase the accuracy and reliability of the scoring results, four members of the study carried out fieldwork on 16 September 2022; the route can be seen in Figure 3. Point A is the North Melbourne train station, which is currently the closest train station to Arden. The streets here will provide quick and convenient walking and cycling connections between key spaces such as the Macaulay, Arden, and North Melbourne train stations, as well as open spaces within and outside of Arden. Point B is the new Arden station, due to open in 2025, which will accelerate Arden’s emergence as the centre of the district and assist in achieving the target of at least 60% public transport, 30% walking and cycling, and no more than 10% private vehicles. Points C, D, and E are the North Melbourne Recreation Reserve, underground storage tanks, and temporary parking, respectively. Flows from Melbourne’s inner northwest have an impact on the Moonee Ponds Creek (point F). The current drain-age system is not designed to handle this amount of stormwater, resulting in dangerous flood depths. According to Figure 4, in the region’s draught vision framework, a sizable portion of land will be at risk of flooding, which will undoubtedly become more severe if sea levels rise by 2.7 m, as described in the worst-case scenario [51]. Point G and point H are outside the reach of the Arden Structure Plan; however, it is worth noting that the residential houses in these areas are also vulnerable to flooding.
The scholars scored the Arden structural plan with reference to the scoring used in previous studies [52,53]. The evaluation of the quality of the Arden Structure Plan was conducted by scoring the indicators on a scale between 0 and 2. A score of ‘0’ was given if there was no evidence or description of the indicator throughout the plan. An indicator was given a score of ‘1’ if the indicator was inferred or mentioned but not in detail, while an indicator was given a score of ‘2’ when it was fully mentioned and discussed in detail. Each factor (AAA) was equally important and had the same weight. The scoring protocol was implemented with comments on each indicator to justify the coding decision. In addition, an assistant professor from the University of Melbourne was invited to validate the scoring protocol to ensure professionalism and reduce the influence of subjective factors by averaging the results.”
- In response to your comments, we removed these figures.
Once again, thank you very much for your constructive comments and suggestions which would help us both in English and in depth to improve the quality of the paper.
Kind regards,
The authors.

Author Response
Dear Editor Zhou and reviewers,
Thank you for your letter and the reviewers’ comments on our manuscript entitled " The Effectiveness of Local Governments’ Policies in Response to Climate Change: An Evaluation of Structure Planning in Arden, Melbourne " (ID: ijerph-2123020). Those comments are very helpful for revising and improving our paper, as well as the important guiding significance to another research. We have studied the comments carefully and made corrections which we hope meet with approval. The main corrections are in the manuscript and the responds to the reviewers’ comments are as follows (the replies are highlighted in blue).
Reviewer 2:
The manuscript is well planned and supported by some researchers of previous research on a similar field of study. The study critically evaluates the structural planning of Arden and proposes strategies for optimization. Special attention is given to the effects of population change and climate change in the year 2100, which is of great importance to the sustainable urban development of Arden, Melbourne. However, I think the paper requires some revision to improve.
- Page 4 Lines 124-145, it is suggested to integrate the content.
- Page 6 Line 189, it is suggested to describe the eight characteristics.
- Page 7 Line 232, the word "Adam" is misspelled.
- Section 3.2.1, this part needs to add the content of “Action”.
- Page 8 Fig 3, the Plan quality assessment framework demonstrates the logic of the whole evaluation, it needs to change the format for more readability and help the reader to join all the dots.
- The secondary headings for section 4 and section 5 could be more descriptive, replacing simple word headings.
Response:
Thank you for your comments. We have made the following modifications.
- In response to your comments, we have integrated these contents. The modifications are as follows.
“2.1. Study Area
Arden, located between Macaulay Road, the Upfield railway line, Moonee Ponds Creek, and Dryburgh Street, is an urban regeneration area in North Melbourne located ap-proximately 2 kilometres from the Melbourne CBD (Figure 1) that comprises 44.6 hectares of industrial land. Arden is situated in a floodplain that is vulnerable to flooding. Ac-cording to the Victorian Planning Authority (2022), Arden is expected to provide approx-imately 34,000 jobs and accommodate approximately 15,000 residents over the next 15 to 30 years and will become an employment and transport precinct and new community for inner Melbourne [44].
The Arden Structure Plan and Planning Scheme Amendment C407 was approved by the Minister for Planning and gazetted on 28 July 2022 under Amendment C407melb to the Melbourne Planning Scheme. Thus, Arden will seize this opportunity to set the stand-ard for best practices in innovation and leadership for sustainable urban renewal, includ-ing through a commitment to achieve net zero emissions for the entire precinct by 2040, in accordance with the Melbourne City Council policy. The Arden Structure Plan aligns with several local plans proposed by the City of Melbourne.”
“2.2.2. Fieldwork and Scoring Protocol
This study invited 15 Melbourne- and Beijing-based urban planning and urban ge-ography scholars, 10 of which held PhDs, including one professor, three associate profes-sors, six assistant professors, and four current master’s students. In order to ensure an ob-jective understanding and real perception of Arden’s current situation, and to increase the accuracy and reliability of the scoring results, four members of the study carried out fieldwork on 16 September 2022; the route can be seen in Figure 3. Point A is the North Melbourne train station, which is currently the closest train station to Arden. The streets here will provide quick and convenient walking and cycling connections between key spaces such as the Macaulay, Arden, and North Melbourne train stations, as well as open spaces within and outside of Arden. Point B is the new Arden station, due to open in 2025, which will accelerate Arden’s emergence as the centre of the district and assist in achiev-ing the target of at least 60% public transport, 30% walking and cycling, and no more than 10% private vehicles. Points C, D, and E are the North Melbourne Recreation Reserve, un-derground storage tanks, and temporary parking, respectively. Flows from Melbourne’s inner northwest have an impact on the Moonee Ponds Creek (point F). The current drain-age system is not designed to handle this amount of stormwater, resulting in dangerous flood depths. According to Figure 4, in the region’s draught vision framework, a sizable portion of land will be at risk of flooding, which will undoubtedly become more severe if sea levels rise by 2.7 m, as described in the worst-case scenario [51]. Point G and point H are outside the reach of the Arden Structure Plan; however, it is worth noting that the resi-dential houses in these areas are also vulnerable to flooding.”
- In response to your comments, we added the eight characteristics. The modifications are as follows.
“Berke et al. expanded these three characteristics to eight characteristics, including fact base, goals, policies, implementation, monitoring and evaluation, inter-organizational coordination, participation, and plan organization and presentation [24].”
- In response to your comments, we have corrected the spelling of Arden.
- In response to your comments, we added the content of “Action”. The modifications are as follows.
“In addition, referring to Hurlimann et al. [41], the climate change assessment framework used here consists of three parts: mitigation, adaptation, and integration. It focuses on actions to address climate change in the Arden Structure Plan and highlights the considerations at the state level and on a smaller scale. The actions component of this framework identifies appropriate measures and monitoring strategies to demonstrate how the local government will achieve sustainable urban development and reduce the trend of climate change, which include implementation, monitoring, and evaluation as well as inter-organizational coordination and participation.”
- In response to your comments, we have redrawn the figure to make it more readable. The modifications are as follows.
- In response to your comments, we have modified the secondary headings. Due to a chapter adjustment, Section 4 of the original manuscript was revised to be Section 3. Section 5 of the original manuscript was proposed to be merged with the discussion section eliminating the secondary heading. The modifications are as follows.
“3.2. Quality of Awareness
3.3. Quality of Analysis
3.4. Quality of Action”
Once again, thank you very much for your constructive comments and suggestions which would help us both in English and in depth to improve the quality of the paper.
Kind regards,
The authors.

Reviewer 3 Report
The introduction section failed in deliver clear scientific problem, or aims of the study, in a systematic way. The section of Site and Policy Context is not necessary, they should be compressed extensively and deliver necessary information together with a section of study area. With In depth and systematic discussion should be reorganized.
Abstract: line 22-24, what is the relationship between the sentence ‘The new railway station … of land’ and the content before and after this sentence?
Line 42-43, is there a bracket without Times New Roman?
Line 54, update the population to the newest one.
Line 73, what is guideline v?
Line 82, glasshouse gas emission?
Line 94, I can not understand the meaning of Physical context, do you mean the study area?
Line 95-145, too many contents which is not closely related to the paper.
Line 146-174, compress these contents as suggested previously.
Line 176-228, these contexts should be moved to section of Introduction.
Line 258, the figure is not very clear, use higher resolution. In addition, the text in the figure is too small.
Line 268-385, with only 5 references in the discussion section will inevitably restrict the depth and systematic of the discussion section.
Line 386-502, the provided recommendations is not based on quantitative analysis, actually, the paper itself is not based on necessary quantitative analysis, so I am not convinced by the method and corresponding results.
Author Response
Dear Editor Zhou and reviewers,
Thank you for your letter and the reviewers’ comments on our manuscript entitled " The Effectiveness of Local Governments’ Policies in Response to Climate Change: An Evaluation of Structure Planning in Arden, Melbourne " (ID: ijerph-2123020). Those comments are very helpful for revising and improving our paper, as well as the important guiding significance to another research. We have studied the comments carefully and made corrections which we hope meet with approval. The main corrections are in the manuscript and the responds to the reviewers’ comments are as follows (the replies are highlighted in blue).
Reviewer 3:
- The introduction section failed in deliver clear scientific problem, or aims of the study, in a systematic way. The section of Site and Policy Context is not necessary, they should be compressed extensively and deliver necessary information together with a section of study area. With In depth and systematic discussion should be reorganized.
- Abstract: line 22-24, what is the relationship between the sentence ‘The new railway station … of land’ and the content before and after this sentence?
- Line 42-43, is there a bracket without Times New Roman?
- Line 54, update the population to the newest one.
- Line 73, what is guideline v?
- Line 82, glasshouse gas emission?
- Line 94, I can not understand the meaning of Physical context, do you mean the study area?
- Line 95-145, too many contents which is not closely related to the paper.
- Line 146-174, compress these contents as suggested previously.
- Line 176-228, these contexts should be moved to section of Introduction.
- Line 258, the figure is not very clear, use higher resolution. In addition, the text in the figure is too small.
- Line 268-385, with only 5 references in the discussion section will inevitably restrict the depth and systematic of the discussion section.
- Line 386-502, the provided recommendations is not based on quantitative analysis, actually, the paper itself is not based on necessary quantitative analysis, so I am not convinced by the method and corresponding results.
Response:
Thank you for your comments. We have made the following modifications.
- In response to your comments, we have rewritten the introduction and study area:
“1. Introduction
Structure planning is a type of spatial planning and is a part of urban planning practices in the United Kingdom and Australia [1]. In Australia, structure plans are commonly prepared at the subregional, district, and local levels under strategic planning. Structure plans set the direction of development in urban sub-regions and play an important role in the implementation and effectiveness of strategic policy objectives. They also make provisions for changing community needs and set out how they should be managed by local government or councils. They guide major changes in land use, transportation integration, built form, public space, and cultural transmission, working together to achieve the eco-nomic, social, and environmental objectives of the area [2,3].
In the last hundred years, the world has experienced significant changes, mainly characterised by warming, with widespread and far-reaching effects [4-6]. In Australia, particularly in metropolitan areas such as Sydney and Melbourne, the pressures of population growth and the effects of climate change compound the problems of environmental degradation, high resource consumption, and the loss of development space [7,8]. The United Nations Environment Programme (2021) claims that as emissions pledges be-tween nations increase in size and complexity, climate change becomes a more serious issue [9]. Additionally, national governments are unable to make a significant contribution towards effectively reducing greenhouse gas emissions (for example, by repealing Australia’s carbon tax). As a result, local governments play a crucial role in directing sustainable development. As of 2022, the population of Melbourne is about 5 million, and the population density in the city is about 500 residents per square kilometre. In fact, around 75% of the population of Victoria lives in Melbourne. Melbourne is not only a populated city, but it is also expanding quickly. Melbourne is expected to overtake Sydney to become Australia’s largest city in about 25 years if it continues to grow at this rate [10]. The structure plan must take on additional functions not only to ensure the economic and social vitality of the area, but also to develop a range of tools to help local governments create and manage their plans to make them more resilient to climate change and increase the liveability of these regions [11].
Given that planning decisions have a decades-long impact on both the natural and built environments, which is sometimes irreversible [12], conducting a programme assessment is a critical step in ensuring that these plans produce the desired results [13,14]. Planning assessments can help to increase the accountability of local governments and public institutions and inform decision makers on ways to address policy issues [15], ensuring that the planning is serious and scientific [16]; in addition, they also strengthen public confidence in planning decisions [17,18]. Scholars have developed analytical methods and conducted empirical studies to assess the quality of planning [19,20]. The question of whether planning assessments should focus on ‘ex ante (or a priori)’, ‘on going’, or ‘and ex post’ has been widely discussed by scholars in various countries [21-25]. Although quantitative indicators play a pivotal role, the joint assessment of qualitative description means is comprehensive, especially for those planning implementation goals that cannot be directly quantified, where qualitative indicator description is also essential [26]. In the 1980s, Canadian scholar Hok-Lin Leung proposed the S-CAD (subjectivity, consistency, adequacy, and dependency) public policy evaluation method, which helps policy-making participants develop and analyse their own policy recommendations as well as evaluate and resolve their differences with other participants ([27-29]. Many governments have also established planning evaluation systems and applied planning evaluation methods in practical work. The UK represents the Western country with the most complete planning evaluation system, and this system was eventually adopted by the EU, forming a regional planning policy evaluation system under the EU framework [30]. Port-land’s Comprehensive Plan Assessment Plan, based on an analysis of the implementation environment and urban development trends of Portland’s 1980 master plan, focuses on assessing the timeliness of the implementation of the plan and provides a systematic assessment of the master plan in seven areas: economic development, environmental protection, housing planning, infrastructure planning, sustainable development, transportation, and urban form [31]. Hong Kong’s Territorial Development Strategy Review focuses on assessing the forward-looking nature of planning, particularly in terms of assessing Hong Kong’s competitiveness in the global urban system [32,33].
Sustainable development and combating climate change are currently some of the most topical issues in urban research and are among the core objectives of urban development. Planning assessment methodologies that address sustainable urban development, climate change, and the maintenance of infrastructure support and ecological and green spaces have been developed, for example, the Urban Sustainability Assessment Framework and the Climate Change Assessment Framework. Holden et al. suggested that urban sustainability should be based on three imperatives: satisfying human needs, ensuring social equity, and respecting environmental limits, identified by six key themes [34]. This emphasizes the primary objective and proposes quantifiable indicators, which makes this definition more forward-looking and feasible both at the state level and municipality levels. Although the neighbourhood is considered the most appropriate spatial scale for improving sustainable built environments [35], bringing sustainability to small neighbourhoods has been acknowledged as a challenge for developers and local municipalities due to a lack of institutional and financial capacities. Compared with larger communities, smaller neighbourhoods are limited in their ability to adopt sustainable practices. Neighbourhood sustainability assessment (NSA) tools have been introduced to measure the effectiveness of sustainability development at this spatial scale [36], including LEED (Leadership in Energy and Environmental Design), BREEAM (Building Re-search Establishment Environmental Assessment Method), and ITACA (Italian Institute for Innovation and Transparency in Procurement and Environmental Compatibility). Haider et al. developed a sustainability assessment framework for a small urban neighbourhood based on the analysis of several NSA tools [37]. These NSA tools are considered reliable in assessing the potential for sustainable development in precincts and achieving sustainability goals [38]. According to this feature, sustainability assessment tools can al-so be used to assess the degree of consideration of sustainable development in specific strategic plan documents. The consideration of climate change in municipalities’ plans provides a crucial opportunity to address this global issue, as cities are recognized as major drivers in addressing climate change actions. Larsen and Kørnøv presented three over-all approaches to address climate change: mitigation, adaptation, and baseline adaptation [39]. Grafakos et al. indicated the significance of integrating mitigation and adaptation for a global transition to respond to climate change [40]. This can help avoid conflict between different policies and bring far-reaching benefits to enhance collaboration. Mitigation is considered as a global-scale issue. However, with further research and the advancement of strategic climate change plans, there is a growing awareness of the impact of bottom-up behaviour on mitigation action. Thus, it is important to consider mitigation at both the global level and local municipal levels. In addition, considering the role of adaptation and integration is also crucial in creating climate change assessment frameworks. In order to assess the effectiveness and advancement of these plans in addressing climate change, Hurlimann et al. introduced an assessment focusing on the actions of mitigation, adaptation, and integration [41].
The quality of the plan is a key determinant of its outcome, and a high-quality plan is more likely to produce positive outcomes [42]. The five million residents in Melbourne deal with serious environmental issues, including drought, bushfires, floods, and heat-waves [43] , especially in Arden, which experiences severe flooding. Therefore, it is necessary to critically evaluate the structure plan for the Melbourne urban renewal area of Arden. The government is looking for suggestions on how to better take climate change and urban sustainability into account in the plan and whether the urban renewal project should move forwards. The primary objective of this study was to evaluate the Arden Structure Plan and provide suggestions on how to improve the plan’s urban sustainability and climate change considerations. Secondly, we aimed to offer guidance on whether Arden is a suitable location for Melbourne’s population growth considering the impacts of climate change anticipated to occur by 2100.”
“2.1. Study Area
Arden, located between Macaulay Road, the Upfield railway line, Moonee Ponds Creek, and Dryburgh Street, is an urban regeneration area in North Melbourne located ap-proximately 2 kilometres from the Melbourne CBD (Figure 1) that comprises 44.6 hectares of industrial land. Arden is situated in a floodplain that is vulnerable to flooding. Ac-cording to the Victorian Planning Authority (2022), Arden is expected to provide approximately 34,000 jobs and accommodate approximately 15,000 residents over the next 15 to 30 years and will become an employment and transport precinct and new community for inner Melbourne [44].
The Arden Structure Plan and Planning Scheme Amendment C407 was approved by the Minister for Planning and gazetted on 28 July 2022 under Amendment C407melb to the Melbourne Planning Scheme. Thus, Arden will seize this opportunity to set the standard for best practices in innovation and leadership for sustainable urban renewal, including through a commitment to achieve net zero emissions for the entire precinct by 2040, in accordance with the Melbourne City Council policy. The Arden Structure Plan aligns with several local plans proposed by the City of Melbourne.
Figure 1. Arden Location Map”
- In response to your comments, we have rewritten the abstract:
“Abstract: It is widely acknowledged that climate change has caused serious environmental issues, including drought, bushfires, floods, and heatwaves, and urban sustainability is currently seriously threatened as a result. Arden is one of the key urban regeneration areas set to experience dramatic residential changes under Melbourne’s development blueprint within the next 20 years. The Arden Structure Plan (2022) outlines specific implementation steps but does not go into detail about the strategies and tactics used to address climate change and urban sustainability. Therefore, there are still problems with the plan, including a lack of information and time-bound development targets, ambiguous public engagement, little focus on urban crime, and insufficient cli-mate change adaptation measures. The plan also considers affordable housing, a mixed-use development pattern that will significantly decrease environmental harm, and active transportation options, primarily walking and bicycling. Considering climate change, this plan will make Arden a suitable location for population growth. This paper aims to evaluate the Arden Structure Plan and make recommendations on how to improve the plan’s urban sustainability and climate change considerations. Furthermore, it provides guidance on whether Arden is a suitable location for Melbourne’s population growth in light of the climate change impacts anticipated to occur by 2100.”
- Done.
- Done.
- We are sorry that was a clerical error.
- We are sorry that was a clerical error. It should be “greenhouse gas”.
- In response to your comments, we have compressed extensively the section of Site and Policy Context, and we rewritten the section of study area. We have written the section:
“2.1. Study Area
Arden, located between Macaulay Road, the Upfield railway line, Moonee Ponds Creek, and Dryburgh Street, is an urban regeneration area in North Melbourne located ap-proximately 2 kilometres from the Melbourne CBD (Figure 1) that comprises 44.6 hectares of industrial land. Arden is situated in a floodplain that is vulnerable to flooding. Ac-cording to the Victorian Planning Authority (2022), Arden is expected to provide approximately 34,000 jobs and accommodate approximately 15,000 residents over the next 15 to 30 years and will become an employment and transport precinct and new community for inner Melbourne [44].
The Arden Structure Plan and Planning Scheme Amendment C407 was approved by the Minister for Planning and gazetted on 28 July 2022 under Amendment C407melb to the Melbourne Planning Scheme. Thus, Arden will seize this opportunity to set the standard for best practices in innovation and leadership for sustainable urban renewal, including through a commitment to achieve net zero emissions for the entire precinct by 2040, in accordance with the Melbourne City Council policy. The Arden Structure Plan aligns with several local plans proposed by the City of Melbourne.”
- Please see the previous comment 7 for a modification of this comment. We have compressed extensively the section of Site and Policy Context, and we rewritten the section of study area.
- Please see the previous comment 7 for a modification of this comment. We have compressed extensively the section of Site and Policy Context, and we rewritten the section of study area.
- We have rewritten the introduction and merged this section with the introduction. Please see the previous comment 1 for a modification of this comment.
- Done.
- We have rewritten the discussion section and added about 20 references. We have written the section:
“4. Discussion
The Arden Structure Plan was published in July 2022. Although we did not assess further planning options, we did assess the planning issues, planning objectives, and strategies that the plan could address [55,56]; thus, this work is similar to an ex ante evaluation. A considerable effort was made to try to predict whether the planning objectives and solutions outlined in the Arden Structure Plan are scientific and relevant in the con-text of climate change and sustainable urban development. In this study, we integrated an assessment of urban sustainability and climate change with an assessment of the quality of the plan, as well as constructed an assessment framework that combined the quantitative and qualitative assessment of not only the urban sustainability and climate change elements of the Arden Structure Plan but also the feasibility and public participation of the plan as a whole. As Baer pointed out, there is no single standard for planning assessment; the established standards only provide a reference for planning assessment practitioners and are not decisive [57]. It is often possible to interpret them systematically by setting corresponding criteria [58]. Therefore, the assessment framework draws on relevant indicators from existing references [24,50] and adapts them to the water-sensitive geographical characteristics of Arden’s floodplain, focusing more on the impact of flooding on the area by setting indicators such as “groundwater pumping”, “percentage of 1-year, 24-h storm runoff for the entire community”, and “distance to critical floodplain/100–1000 years)”. The assessment framework is friendly and flexible for planners in water-sensitive areas and helps planners and decision makers to assess the quality of integrated urban sustainability and climate change plans.
We attempted to make recommendations for aspects of the Arden Structural Planning Assessment that are underperforming. In 2021, the census resident population of Arden was 413, living in 224 dwellings with an average household size of 2.11 [59]. Melbourne’s population will grow to 30 million by 2100 [59]. Arden has an abundance of land, and the new railway station will provide opportunities for local development and population growth. The plan takes into account the provision of 6% affordable housing for low- and middle-income people, which will also avoid gentrification caused by population growth and contribute to the sustainability of the city’s population. The Victorian government is aware of the impact of climate change on urban sustainability and has the ambition to address it. However, the presentation of the context and impacts of urban sustainability and climate change should be more regionally specific and provide a clear-er explanation of the concepts, which will help non-specialists to understand the significance and need for relevant planning. The future development of Arden as a regional centre with a new railway station and a large number of buildings and roads with a high heat storage capacity will inevitably result in the area becoming a new heat island in North Melbourne. The plan should recognise this before it happens and propose specific approaches to avoid the formation of a new urban heat island, including strategies for the construction of physical spaces such as buildings and roads, for example, using the ab-sorption and reflection of solar energy by colour to reduce building energy consumption, and for promoting the use of a variety of public transportation options. In addition, it is also important to promote science and technology and to guide people to develop ecological and energy-saving habits in their lives [60]. Different environmental behaviours can be encouraged and punished through a clear system of rewards and penalties. For example, Singapore’s comprehensive eco-accountability system [61] relies on effective government policies, public participation, and transparent information to create a beautiful natural environment and a garden-like city [62].
The lack of analysis in the Arden Structure Plan provides little guidance for the preparation and practice of planning. The plan scored lowest on the analysis factor and lacked specific planning strategies and measures related to urban sustainability and the response to climate change. According to the three ethical imperatives for sustainable urban development [34], in terms of environmental limits, the environmental resources of the city are limited; thus, the development model of the city will increasingly be fine-tuned and tend towards information-based management. The Arden plan lacks a large amount of basic data on the urban environment to support it. Big data technology can be used to collect, collate, and analyse data and information about the urban environment to provide a more effective scientific strategy for the prevention of urban natural disasters. Only with a full understanding of the natural environmental base of the area can the boundaries of development be clarified to avoid encroaching on the environmental resources that should be left to future generations. In term of human needs, the Arden Structure Plan (2022) is in-adequate in terms of crime prevention measures; however, urban safety is one of the most fundamental human needs. Cozens proposed that ‘designing out crime’ represents an important tool to help develop urban sustainability [63]. A city cannot be called sustainable if its residents are concerned about safety [64]. The Arden plan, which only proposes to promote street surveillance by a layer of buildings around the main pavements, is incomplete. Therefore, the government should consider developing products and systems that are more resistant to crime to reduce its impact and also use the built environment to re-duce waste generation, promote liveability, improve quality of life, and reduce opportunities for crime. In terms of social equity, the analysis showed that the ‘Participation by res-idents’ and ‘Embrace and recognize differences’ components both scored zero. Berke et al. stated that a quality plan should include a stakeholder engagement approach [24]. In addition, according to Tan and Artist [65], citizen participation is necessary as public ser-vices are not the same as private sector services. The Arden Structure Plan (2022) clearly lacks a specific approach to enhancing community participation and does not focus on how special groups and vulnerable populations can actually be involved in the development of plans. Sustainable cities must respect human rights and be planned and designed with people in mind. The community should strengthen its connection with residents, publicize and share relevant policy information in a timely manner, and enhance public opinion surveys to ensure that decisions can truly serve and benefit all residents. This will also help foster a sense of responsibility and territorial belonging for the development of Arden, such that residents will feel that everyone has a voice in decision making and that the community operates in an equitable manner.
In 2021, UNEP states that global temperatures will rise by 2.7 °C in 2100, which is well above the Paris Agreement temperature control target. The Arden Structure Plan (2022) proposes net-zero emissions across the precinct by 2040, where all planning decisions and Arden’s active transport options will support this goal. The government seems to be ambitious and prepared to adapt and mitigate the impacts of climate change in Arden. The planned mixed land use pattern will effectively reduce the environmental dam-age caused by urban construction, and active transport options, mainly walking and cy-cling, will reduce environmental pollution. All these measures will make Arden a suitable place for population growth in the context of climate change. Measures to address climate change mainly include mitigation, adaptation, and the integration of the two. Climate change mitigation targets should contain both qualitative and quantitative characteristics, such as specific warming limits and data on greenhouse gas emission reductions, which are lacking in the Arden Structure Plan (2022). According to Albrechts, there is a risk of losing momentum if there are no short-term goals to be achieved [66]. The plan should therefore establish a schedule of specific mitigation targets for different periods to ensure that its vision can be achieved over time. In addition, less attention is given to adaptation than to mitigation in planning. However, the planning should take full account of the particularities of the Arden environment, which is vulnerable to disturbances from multiple sources of flooding, and propose specific measures to adapt to this regional specificity. The plan presents a good vision of water in Arden as a landscape feature through innovative flood management solutions. However, the basic data and measures of the flood protection plan are not detailed enough, and the management of the flood protection stock, the emergency management mechanism, the construction of flood protection works, and the delineation of safety zones need to be presented more clearly. Addressing climate change is not the sole responsibility of one sector; mitigation and adaptation measures need to work together, and all relevant sectors should share the impacts of climate change. The 3S technique was used to analyse floodplains and water ecological zoning to draw conclusions about water ecological sensitivity in terms of setting up soil and water conservation forests between built-up areas and rivers and lakes to achieve water retention and filtration and to guide site layout [67]. Flooding can have a direct and indirect impact on transport facilities in Arden, and heavy rainfall also affects the way people move around. In the face of emergency climate change situations, the transport department and the meteorological department should liaise in a timely manner to take emergency warning measures and inform residents in advance of appropriate travel options to mitigate the effects of special climate change in a timely manner. At the same time, the routine maintenance of transport facilities should be carried out to adapt their materials and permeability to climate change. For example, Stockholm deals with heavy rainfall by using green roofs in buildings, which are porous surfaces for on-site infiltration that treat rain-water for new water storage technologies and green spaces, rather than discharging it into the sewage system [68]. In addition, integrating green and grey infrastructure (traditional municipal facilities) into one sustainable stormwater management system can address land-use constraints, fulfil the stormwater function of green infrastructure [69], enhance urban plaza landscaping, and increase public accessibility to the water. This will help Arden mitigate and adapt to the impact of extreme weather on the urban environment and people’s lives and reduce property damage.
The most important area for improvement in the action section is the low-scoring monitoring and evaluation. Climate change is an unpredictable process, and planning is dynamic in nature. Big data is not only used for climate change observation, monitoring, early warning, and trend prediction, but also for urban disaster identification and assessment, and public engagement [70,71]. Big data can provide important support for the science of planning, decision making, implementation, and evaluation, both in terms of research paradigms and technical methods and has great potential for the development of climate change adaptation planning strategies [72,73]. The Arden Structure Plan (2022) needs to strengthen the way it is monitored and evaluated in order to obtain real-time data and adjust its content in a timely manner. The continuous tracking of implemented activities will also help to determine the extent to which targets are being met [74].”
- We have revised the entire article and added appropriate content to make it more convincing. We have combined the discussion and suggestions and rewritten this section. Please see the previous comment 12 for a modification of this comment.
Once again, thank you very much for your constructive comments and suggestions which would help us both in English and in depth to improve the quality of the paper.
Kind regards,
The authors.

Reviewer 4 Report
This paper addresses an evaluation of Structure Plan of Arden, Melbourne. Overall, the manuscript is well organized and its presentation is good. However, some minor issues still need to be improved:
(1) There are some minor errors in writing, such as line 25,31,53 and so on. Please check the whole article.
(2) Who scored the plan quality evaluation of the Adren Structure Plan about the part of scoring protocol in this paper, whether other experts participated in it.
Author Response
Dear Editor Zhou and reviewers,
Thank you for your letter and the reviewers’ comments on our manuscript entitled " The Effectiveness of Local Governments’ Policies in Response to Climate Change: An Evaluation of Structure Planning in Arden, Melbourne " (ID: ijerph-2123020). Those comments are very helpful for revising and improving our paper, as well as the important guiding significance to another research. We have studied the comments carefully and made corrections which we hope meet with approval. The main corrections are in the manuscript and the responds to the reviewers’ comments are as follows (the replies are highlighted in blue).
Reviewer 4:
This paper addresses an evaluation of Structure Plan of Arden, Melbourne. Overall, the manuscript is well organized and its presentation is good. However, some minor issues still need to be improved:
(1) There are some minor errors in writing, such as line 25,31,53 and so on. Please check the whole article.
(2) Who scored the plan quality evaluation of the Adren Structure Plan about the part of scoring protocol in this paper, whether other experts participated in it.
Response:
Thank you for your comments. We have made the following modifications.
(1)In response to your comments, we have modified the minor errors and check the whole article. In addition, we have used the language editing by MDPI.
(2)In response to your comments, we elaborate on this in the Methods section. The modifications are as follows.
“2.2.2. Fieldwork and Scoring Protocol
This study invited 15 Melbourne- and Beijing-based urban planning and urban ge-ography scholars, 10 of which held PhDs, including one professor, three associate profes-sors, six assistant professors, and four current master’s students. In order to ensure an ob-jective understanding and real perception of Arden’s current situation, and to increase the accuracy and reliability of the scoring results, four members of the study carried out fieldwork on 16 September 2022; the route can be seen in Figure 3. Point A is the North Melbourne train station, which is currently the closest train station to Arden. The streets here will provide quick and convenient walking and cycling connections between key spaces such as the Macaulay, Arden, and North Melbourne train stations, as well as open spaces within and outside of Arden. Point B is the new Arden station, due to open in 2025, which will accelerate Arden’s emergence as the centre of the district and assist in achiev-ing the target of at least 60% public transport, 30% walking and cycling, and no more than 10% private vehicles. Points C, D, and E are the North Melbourne Recreation Reserve, un-derground storage tanks, and temporary parking, respectively. Flows from Melbourne’s inner northwest have an impact on the Moonee Ponds Creek (point F). The current drain-age system is not designed to handle this amount of stormwater, resulting in dangerous flood depths. According to Figure 4, in the region’s draught vision framework, a sizable portion of land will be at risk of flooding, which will undoubtedly become more severe if sea levels rise by 2.7 m, as described in the worst-case scenario [51]. Point G and point H are outside the reach of the Arden Structure Plan; however, it is worth noting that the resi-dential houses in these areas are also vulnerable to flooding.
The scholars scored the Arden structural plan with reference to the scoring used in previous studies [52,53]. The evaluation of the quality of the Arden Structure Plan was conducted by scoring the indicators on a scale between 0 and 2. A score of ‘0’ was given if there was no evidence or description of the indicator throughout the plan. An indicator was given a score of ‘1’ if the indicator was inferred or mentioned but not in detail, while an indicator was given a score of ‘2’ when it was fully mentioned and discussed in detail. Each factor (AAA) was equally important and had the same weight. The scoring protocol was implemented with comments on each indicator to justify the coding decision. In ad-dition, an assistant professor from the University of Melbourne was invited to validate the scoring protocol to ensure professionalism and reduce the influence of subjective factors by averaging the results.”
Once again, thank you very much for your constructive comments and suggestions which would help us both in English and in depth to improve the quality of the paper.
Kind regards,
The authors.

Round 2
Reviewer 1 Report
Several changes were identified in the manuscript titled “the effectiveness of local governments’ policies in response to climate change: an evaluation of structure planning in Arden, Melbourne” However, some minor remarks are provided to improve this manuscript:
Section 2.2.1. “… the AAA approach was adapted to form a new plan quality assessment protocol focusing on these two themes”. The authors must show evidence of this protocol.
Section 2.2.2. “This study invited 15 Melbourne- and Beijing-based urban planning and urban geography scholars, 10 of which held PhDs, including one professor, three associate professors, six assistant professors, and four current master’s students.” Was there any trend in the scoring results? Maybe, urban scholars provide different values than those provided by PhD or professors. An in-depth discussion of the results is required.
Section 3.1. “A complete list of indicators and their evaluation is presented in Appendix 1.”. I could not access to Appendix 1. Supplementary materials only included the responses to reviewers.
Section 4. “In this study, we integrated an assessment of urban sustainability and climate change with an assessment of the quality of the plan, as well as constructed an assessment framework that combined the quantitative and qualitative assessment of not only the urban sustainability and climate change elements of the Arden Structure Plan but also the feasibility and public participation of the plan as a whole.” The authors must show evidence of this integrated assessment. This evidence should be included in the supplementary materials and described before the scoring sections of this manuscript.
Section 4. “It is often possible to interpret them systematically by setting corresponding criteria”. I agree. This manuscript must describe in detail this interpretation and its corresponding criteria.
Author Response
Dear Editor Chen and reviewers,
Thank you for your letter and the reviewers’ comments on our manuscript entitled " The Effectiveness of Local Governments’ Policies in Response to Climate Change: An Evaluation of Structure Planning in Arden, Melbourne " (ID: ijerph-2123020). Those comments are very helpful for revising and improving our paper. We have studied the comments carefully and made corrections which we hope meet with approval. The main corrections are in the manuscript and the responds to the reviewers’ comments are as follows (the replies are highlighted in blue).
Reviewer 1:
Several changes were identified in the manuscript titled “the effectiveness of local governments’ policies in response to climate change: an evaluation of structure planning in Arden, Melbourne” However, some minor remarks are provided to improve this manuscript:
1.Section 2.2.1. “… the AAA approach was adapted to form a new plan quality assessment protocol focusing on these two themes”. The authors must show evidence of this protocol.
2.Section 2.2.2. “This study invited 15 Melbourne- and Beijing-based urban planning and urban geography scholars, 10 of which held PhDs, including one professor, three associate professors, six assistant professors, and four current master’s students.” Was there any trend in the scoring results? Maybe, urban scholars provide different values than those provided by PhD or professors. An in-depth discussion of the results is required.
3.Section 3.1. “A complete list of indicators and their evaluation is presented in Appendix 1.”. I could not access to Appendix 1. Supplementary materials only included the responses to reviewers.
4.Section 4. “In this study, we integrated an assessment of urban sustainability and climate change with an assessment of the quality of the plan, as well as constructed an assessment framework that combined the quantitative and qualitative assessment of not only the urban sustainability and climate change elements of the Arden Structure Plan but also the feasibility and public participation of the plan as a whole.” The authors must show evidence of this integrated assessment. This evidence should be included in the supplementary materials and described before the scoring sections of this manuscript.
5.Section 4. “It is often possible to interpret them systematically by setting corresponding criteria”. I agree. This manuscript must describe in detail this interpretation and its corresponding criteria.
Response:
- In response to your comments, we have added the Plan quality assessment framework include assessment indicators and criteria description after the references. We describe the scoring rules in more detail in the scoring protocol section. The details are as follows.
“Appendix 1
Plan quality assessment framework |
||||||
Factors |
Category |
Indicator |
Objectives |
Description/Criteria |
Score |
Comments |
Awareness |
Context |
Context of Urban Sustainability and Climate change |
—— |
Does the plan include urban sustainability and climate change issues in the context of the site? |
1 |
The plan mentions the history, current development and shortcomings of Arden in the context, partly in relation to urban sustainability, but is vague. |
Concept |
Concept of Urban Sustainability and Climate change |
—— |
Does the plan include a description of the causes of climate change and the knowledge of urban sustainability? |
1 |
This plan does not specifically describe sustainability and climate change, but the strategy covers both two perspectives. |
|
Impacts |
Present impacts of Urban Sustainability and Climate change |
—— |
Does the plan include a discussion of the general impacts of climate change and urban sustainability? |
1 |
The plan mentions relevant content but lacks systematic descriptions. |
|
Targets |
Targets of Urban Sustainability and Climate change |
—— |
Does the plan include at least one goal related to climate change and urban sustainability? |
2 |
The plan proposes to embed sustainable change and some strategies about climate change. |
|
Analysis |
Urban Sustainability |
Environmental limits |
Natural Land Protection |
Nearby drinking water supply or source |
0 |
Arden is considering for an alternative water supply for indoor and outdoor non‐potable uses in Arden. Alternative water sources under investigation include local storm water harvesting and sewer mining. |
Nearby wastewater system |
0 |
The plan only Mentions the natural water management. |
||||
Nearby transportation infrastructure |
2 |
The plan propose high capacity public transport capable options and transport links connecting. |
||||
Brownfield area |
1 |
The plan mentions the management of land contamination |
||||
Existence of imperiled species |
0 |
This criterion is not mentioned, but should be considered. |
||||
Distance from wetlands (ft.) |
1 |
The plan describes that Arden is formerly a low‐lying wetland. |
||||
Distance from water bodies (ft.) |
1 |
Shown in Vision plan. |
||||
% of area (more than 15% slope) protected |
0 |
This criterion is not mentioned, but should be considered. |
||||
Residential dwelling units per acre of buildable land |
0 |
This criterion is not mentioned, but should be considered. |
||||
Use corridors etc. to form networks with the surroundings (Continuity of green space through rows of trees, shrubbery etc.). |
1 |
The plan proposes that green links and green streets will be continue to transform through extensive tree planting. |
||||
Environment building, considering natural flora in the surroundings. |
0 |
This criterion is not mentioned, but should be considered. |
||||
% of urban area protected as agriculture area for fresh food |
0 |
This criterion is not mentioned, but should be considered. |
||||
Ground water pumping |
2 |
Arden currently has five existing pumps and the Arden Flood Management Strategy proposes to upgrade the pumping stations and pipelines.The URCRS will also collect funds from developers to build pump stations. |
||||
% of 1 year, 24 hour storm runoff volume across the community |
0 |
No information in the plan, only a maximum storage capacity of 40 ML is mentioned. |
||||
% paved area in the designated area |
1 |
Objective 17 mentions that pedestrian priority areas will be enhanced by textured pavement changes, but there is no specific percentage. |
||||
Reuse of construction site soil |
2 |
Objective 3 and 6 refer to the adaptive reuse of built heritage sites to adapt to changing community needs |
||||
Combine utility lines to minimize land disturbance |
1 |
The plan proposes combining residential development with commercial, which means that some use routes will be combined, but this is not explicitly stated. Objective 4 proposes combining some blocks according to use needs, but still ensuring a connected, safe road network for smaller streets and driveways. |
||||
Consideration of scenic appearance |
2 |
Arden will be shaped by typical architectural forms. the architectural form of Laurens Streets complements and builds on the character of North Melbourne and will create a visual transition. |
||||
% of buildable land protected as greenspace |
0 |
This criterion is not mentioned, but should be considered. |
||||
Resource Efficiency |
% of boundary connected or adjacent to developed urban area |
1 |
The plan mentions the connection of boundary road with other North Melbourne areas. |
|||
% of Single family dwellings |
0 |
This criterion is not mentioned, but should be considered. |
||||
% of building sq. footage for which superstructure construction is employing BMPs1 for durability and moisture management |
0 |
This criterion is not mentioned, but should be considered. |
||||
% of building sq. footage for which substructure is employing BMPs for durability and moisture control |
0 |
This criterion is not mentioned, but should be considered. |
||||
% of building water use reduction below baseline buildings through the use of water efficient fixtures |
1 |
In the plan, water reuse and efficient energy consumption are embedded in the building design, but no specific data is provided. |
||||
Mutual use of centralized rainwater storage tanks |
2 |
Objective 20 mentions that new development can manage stormwater runoff from centralised wetlands through the provision of rainwater tanks. |
||||
% of landscape area fulfilling the criteria for efficient irrigation system |
2 |
Objective 14 proposes a minimum tree canopy cover of 40% by 2040 under Melbourne's Urban Forest Strategy. |
||||
% of wastewater reused |
1 |
Arden Vision 2018 only mentions the reuse of greywater & stromwater, percentage data is missing. |
||||
% capacity of alternative treatment system beyond the initial design |
1 |
Objective 10 refers to the early provision of alternative waste infrastructure on a regional scale to achieve sustainable development, but no specific percentage figures are available. |
||||
Considerations to minimize water leakage in water supply systems |
0 |
Information is lacking in the plan, but this is an important element to be considered. |
||||
% of Building sq. footage employing the use of energy efficient fixtures and technologies |
1 |
Objective 12 refers to ensuring that new technologies are introduced into buildings, such as energy trading technology and electrification of homes, but there is no data on percentages. |
||||
% of energy from fossil fuels (natural gas, gasoline etc.)/hydro or other water intensive energy sources |
1 |
Stategyc10.4 refers to the provision of fossil fuel free infrastructure for most areas, supporting net-zero carbon emission ambitions, but lacks basic data on the high energy consumption still in use. |
||||
% of energy from renewable sources |
2 |
Strategy 10.1 sets out the ambition for the region to have access to 100% renewable energy and to explore procurement opportunities to address the energy needs of the surrounding communities. |
||||
Centralized storage facilities for waste collection, facilities to reduce volume and employ composting |
2 |
Strategy 10.2 refers to promoting and encouraging the centralisation and sharing of waste management sites in order to reduce freight and waste vehicle emissions. |
||||
Use of locally-produced materials for building cladding, paving and other materials. |
2 |
Strategy 13.3 proposes to encourage the use of locally sourced building materials. |
||||
Reducing the generation of construction by-products |
2 |
Strategy 13.3 proposes to encourage the use of low carbon content feedstocks or production, where possible. |
||||
Sorting and recycling of construction by-products |
2 |
Strategy 13.3 proposes to encourage composting, recycling and reuse of construction materials. |
||||
Use of recycled products (recycled aggregate, blast furnace cement, electric furnace steel etc.) |
2 |
Strategy 13.3 proposes to encourage the use of renewable raw materials. |
||||
Use of materials with low climate change impact (blended cement etc.) |
2 |
Strategy 13.3 proposes to encourage the use of non-hazardous building materials. |
||||
Use of timber from sustainable forestry |
1 |
Strategy 13.3 indirectly expresses the use of sustainable materials, but does not mention sustainable forestry. |
||||
Environmental Quality |
Urban form that secures continuity in airflow within open spaces |
2 |
The plan mentions to design to ameliorates the effects of unsafe wind conditions, and deliver comfortable wind conditions in the public realm for walking, sitting or standing. |
|||
Counter measures against strong winds for regional characteristics |
0 |
The plan mentions wind control but no measures and seems not related to strong wind. |
||||
% of hardscape with surface reflectance index (SRI)>30 |
1 |
The plan requires all new buildings to use materials that minimise the urban heat island effect with a standard that at least 75 per cent of total project site areas should comprise of building or landscaping elements that increase the solar reflectance of the site. |
||||
% of building sq. footage for which BMPs are followed to ensure better indoor environmental quality |
0 |
This criterion is not mentioned, but should be considered. |
||||
Horizontal shaded area ratio from medium and tall trees, e.g., piloti, eaves, pergolas etc. |
2 |
The plan proposes that a target of 40 per cent canopy coverage should be achieved in the public realm over time. |
||||
% of residential area within 300 m walk from green space (>= 2 hectares in size) |
1 |
The service distance of public space is considered in the plan, but there is no specific data about residential area. |
||||
% of Building sq. footage designed for high performance building envelope |
2 |
Strategy 14.2 requires that new buildings reduce materials that lead to the urban heat island effect and that at least 75% of the buildings or landscape elements on the site should have increased solar reflectance. |
||||
Reduction of waste heat |
2 |
Objective 14 proposes that Melbourne is actively tackling this issue by 'greening' the public realm, cooling the environment by providing shade and transferring heat in the landscape. |
||||
% decrease in impervious area |
1 |
The urban water cycle & integrated water management mentions that in Arden, impervious surfaces have significantly altered local water systems and reduced evaporation, which Melbourne will mitigate through greening of public areas, but no specific data is available. |
||||
% of project pavements built with minimum albedo of 0.3 or permeable pavements/pavers |
0 |
This criterion is not mentioned, but should be considered. |
||||
% of streets with trees on both sides with avg. of 40 ft. spacing |
1 |
Strategy 22.3 proposes a tree cover of 40%, but does not address average spacing. |
||||
Distance from critical flood plain (100-1000 years) |
0 |
This criterion is not mentioned, but should be considered. |
||||
% of green space area to reduce heat island effect |
2 |
Th plan require all new buildings to meet a standard of 40 per cent total surface area as green cover; achieves a minimum canopy cover of 40 per cent by 2040. |
||||
Human Needs |
Public Mobility and Accessibility |
% of residential area within 600 m of stores, banks and administrative buildings |
1 |
There is a land function zoning map in the plan, but no specific data. Sub-precincts |
||
% of residential area within 600 m of medical and welfare facilities |
1 |
There is a land function zoning map in the plan, but no specific data. Sub-precincts |
||||
% of residential area within 600 m of educational and cultural facilities |
1 |
There is a land function zoning map in the plan, but no specific data. Sub-precincts |
||||
% of buildings frontage with 25 ft. or less setback |
1 |
Strategy 5.1 proposes controls for setbacks, but no specific data are available. |
||||
Measures taken to manage the traffic demand |
2 |
The plan proposes that Arden will prioritise people walking, cycling and using public transport to meet their daily needs. A network of walkable streets and protected cycle paths is planned. |
||||
Measures taken to calm traffic in urban centers |
2 |
The plan describes that Arden Central-Innovation to be activated by new metro station. |
||||
% of parking space reduction below local minimum or ECC criteria |
1 |
Objective 18 refers to controlling the number of parking spaces and encouraging public transport through parking overlays and other planning controls. However, specific data is missing. |
||||
Provide adequate bike lanes for roads with speed limit 35 mph or over |
1 |
Objective 16 refers to the provision of safe, direct and connected protected cycle paths. |
||||
Accommodation of bicycles near all primary building entrances |
0 |
Objecctive 17 proposes that the streets of the new station in Arden will be accessible to slow-moving cyclists, but does not specifically mention the entrances and exits of all major buildings. |
||||
% of roads (speed limit> 30 mph) with footpath width more than or equal to 900 mm on both sides |
0 |
This criterion is not mentioned, but should be considered. |
||||
% of bus-stop shelters with all-weather protection (rain, wind, snow) and renewable lighting/heating |
0 |
This criterion is not mentioned, but should be considered. |
||||
Visibility from the surroundings |
2 |
Objective 18 propose that new and existing streets will be pedestrian friendly and provide comfortable, green links between open spaces and public transport routes and enhance the quality of the public realm, so the urban spaces are visible and accessible. |
||||
Ease of approach by people with criminal intent |
0 |
There is no description in the plan about criminal intent. |
||||
Provision of street furniture, signs etc. |
1 |
The plan provide a providing a complete list of infrastructure items, but it does not mention street facilities in detail |
||||
Other considerations (lighting, sound etc.) |
2 |
Objective 28 popose to ensure that urban environmet is not unduly impacted by noise, vibration and electromagnetic interference from the adjacent railway corridor, elevated roadway and Metro Tunnel. |
||||
Provision of road width (at least 8m) and formation of a network of evacuation routes, in 2 directions etc. |
2 |
Objective 4 propose to create a new urban structure for Arden that incorporates a high quality network of connected streets, which has minimum 8m-width lanes. |
||||
Continuation of pre-existing community |
2 |
The plan is about urban regeneration, so it is a optimization of pre-existing comuunity |
||||
Social Well-being |
% of project pavements that reduce tire pavement noise levels below certain limits |
0 |
This criterion is not mentioned, but should be considered. |
|||
Noise and vibration counter measures |
2 |
Objective 28 refers to the state government's efforts to relocate industrial businesses out of the district to ensure that noise disturbance from industrial sites is eliminated. The tunnel design of the metro also includes measures to mitigate vibration and noise disturbance. |
||||
Provision of crime prevention facilities, such as surveillance cameras and guards (surveillance of blind spots) |
2 |
Strategy 17.5 proposes through the use of Crime Prevention Through Environment Design principles to promote the use of open space and the ground floor around major pedestrian routes to maximise personal safety. |
||||
Economic Prosperity |
% of regularly trafficked lanes designed for long-life (>=40 years) |
0 |
No information is mentioned in the plan. |
|||
Efforts to track pavement performance through testing and condition assessment procedures |
1 |
Strategy 17.7 proposes to review and update road speed limits in accordance with Arden's principles for movement, but there are no specific procedures or measures. |
||||
Development of facilities related to local industries and culture |
2 |
Objective1propose to celebrate, protect and interpret Aboriginal cultural values and heritage in the planning, design and curation of Arden |
||||
% of labor from local area |
2 |
Strategy 3.1 propose that facilitate use and development of land in Arden to deliver on the aspiration of approximately 34,000 jobs and around 15,000 residents in the precinct. |
||||
Proportion of total house and land packages delivered to the market at an affordable purchase price (less than 270,000$ - 2008 figure - Australia) for moderate income households. |
2 |
Oblective 23 propose that at least 6% of all new housing in Arden is affordable for very low to moderate income households and delivered as social and affordable housing or shared equity. |
||||
% of land costed at lowest quartile of local market |
0 |
There is nothing in the plan about land costs. |
||||
Social Equity |
Public Mobility and Accessibility |
Realization of barrier free outside spaces for the infirm or handicapped. |
1 |
Strategy 18.2 plan to provide disabled parking. Strategy 25.5 propose to provide diverse and adaptable community facilities to become benificial to people with disablities, but the measures are quite general. |
||
Participation by residents of the designated area in planning processes |
0 |
There is nothing in the plan about the participation by residents. |
||||
Consideration of encounter |
2 |
Objective 21 propose to provide generous, well-designed and accessible open spaces that are diverse and flexible to meet the needs of Arden’s evolving community and visitors to the precinct. |
||||
% of primary entryways facing public spaces |
0 |
This criterion is not mentioned, but should be considered. |
||||
Resource Efficiency |
Reasonable redistribution of resources for vulnerable groups |
1 |
Objective 5 propose to make Arden become a world leading urban renewal and innovation precinct. Urban renewal is one of a way for resources redistribution. |
|||
Social Well-being |
Embrace and recognize differences |
0 |
The plan does not mention the recognition and inclusion of social differences |
|||
Overcome gentrification and residential segregation |
2 |
Objective 23 propose to facilitate inclusive, well-designed, sustainable and accessible housing, with affordable housing for very low to moderate income households and delivered as social and affordable housing or shared equity. |
||||
Climate change |
Mitigation |
—— |
Does the plan address climate change mitigation? |
2 |
The plan promotes climate change mitigation, such as minimising waste production and mitigating the urban heat island effect. |
|
Is a mitigation aim specified(e.g. specific level of warming limit, goals for GHG emission reducetion?) If yes, what level? |
2 |
Objective 9 propose to establish strong environmental governance in Arden to achieve the precinct’s net-zero carbon emissions target by 2040, which is adequate to achieve Pairs Argreement. |
||||
Affordable housing mandates |
2 |
Oblective 23 propose that at least 6% of all new housing in Arden is affordable for very low to moderate income households and delivered as social and affordable housing or shared equity. |
||||
Mixed use zoning to support self-contained communities |
2 |
Obljective 25 propose to ensure timely delivery of high-quality, accessible and integrated community infrastructure to meet the needs of existing and future residents, workers and visitors. |
||||
Transfer of development rights to support transit-oriented conrridors |
1 |
Objective 15 propose to provide space for high capacity public transport capable options and improving transport links connecting Arden into the expanding central city, but it doesn's mention the development rights. |
||||
Mixed use zoning to encourage urban regeneration and infill |
2 |
The plan focuses on the urban renewal of Arden, and encourage efficient and diverse use of land. |
||||
Zoning overlays and form-based codes to achieve desired outcomes such as new urbanism |
2 |
Strategy 5.1 implement built form controls in the planning scheme that respond to key design recommendations. |
||||
Design codes and flexible parking to achive transit-oriented development |
2 |
Strategy 18.1 propose to prepare planning controls that direct the ongoing supply and location of car parking to achieve the 10% mode share target for private vehicles and car parking principles, and then minimise the impact of car parking and associated vehicular movements through Arden. |
||||
Mixed-use zoning to support eco-communities |
1 |
The plan concentrates on the sastainable change in community,including some mesures of eco-communities, but it dose not have a systematic description. |
||||
Code revisions/design guidelines for pedestrian zones |
2 |
Strategy 17.1 propose to facilitate a network of permeable streets and pedestrian links through the precinct for walking and cycling that are considerate of safety and convenience and provide direct access to and from key destinations. |
||||
Code revisions/design guidelines to achive car-free disctricts &/or traffic calming |
1 |
Strategy 17.1 encourage the consolidation of servicing facilities and alternative freight delivery models within the precinct to reduce the number of vehicles entering and circulating, but there is no description of car-free disctrict. |
||||
Adaptation |
—— |
Does the plan address climate change adaptation? |
2 |
Objective 12 propose to measure the performance of the precinct, its buildings and its occupants and be able to adapt to changes in climate, lifestyle and technology in the future. |
||
What level of flooding risk is planned for, and across what time frame? |
0 |
The plan mentions flooding management, but there was no clear planning level or timing. |
||||
Does the approach to flooding consider retreat? |
0 |
The plan does not consider retreat for flooding |
||||
Does the approach to flooding consider accommodate? |
2 |
Strategy 19.4 positively respond to any necessary level changes that are required for drainage purposes between development and the public realm, to accommodate flooding problems. |
||||
Does the approach to flooding consier protect? |
2 |
There is the precinct-wide flood management strategy for Arden, to provide guidance on how development can achieve flood responsive design and good urban design outcomes that facilitates safety, equitable access and universal design. |
||||
Integration of mitigation and Adaptation |
—— |
Is the integration of adaptation and mitigation acknowledged? |
1 |
The plan mentions both adaptation and mitigation, but neither is called one-sided and is not described systematically. |
||
Action |
Implementation |
Implementation Section |
—— |
Does the plan include a separate section that addresses what needs to be done to implement the plan? |
2 |
Section 10 called delivery Arden talks mention the development staging of Arden. |
Monitoring and evaluation |
Monitoring and Evaluation Section |
—— |
Does the plan include a separate section that addresses what needs to be done to monitor and evaluate the plan? |
1 |
The plan just mentions the lead agency. |
|
Inter-organizational coordination |
Vertical and horizontal integration |
—— |
Does the plan include at least one vertical connection to federal, provincial plans and regional plans or at least one horizontal connection with other local plans/programs? |
2 |
The plan has connection with other plans such as Moonee Ponds Creek Strategic Opportunities Plan. |
|
Participation |
Stakeholder and public engagement |
—— |
Does the plan identify the organizations and stakeholders involved in the plan making process and identify the public as part of the plan making process |
1 |
The plan just mentions the lead agency and lacks the public participation |
”
“The scholars scored the Arden structural plan with reference [52,53] to the scoring used in previous studies. The evaluation of the quality of the Arden Structure Plan was conducted by scoring the indicators on a scale between 0 and 2 [50]. A score of ‘0’ was given if there was no evidence or description of the indicator throughout the plan. An indicator was given a score of ‘1’ if the indicator was inferred or mentioned but not in detail, while an indicator was given a score of ‘2’ when it was fully mentioned and discussed in detail. Each factor (AAA) was equally important and had the same weight. The scoring protocol was implemented with comments on each indicator to justify the coding decision. The evaluation protocol contains 14 indicators with a maximum score of 28 points. These indicators are divided according to AAA as follows: awareness (4/14), analysis (6/14) and action (4/14). Scoring is done by category, with a maximum of 8 points for awareness, 12 points for analysis and 8 points for action. In addition, an assistant professor from the University of Melbourne was invited to validate the scoring protocol to ensure professionalism and reduce the influence of subjective factors by averaging the results.”
- In response to your comments, we have added a discussion of the differences among the 15 scoring scholars at the end of the first paragraph of Section 4. In fact, the differences are not very significant, which is one of the strengths of this assessment framework and scoring protocol. However, we also feel that this fact should be described in the manuscript. Thank you very much for the reminder. The details are as follows.
“It is worth noting that there is no significant difference in scoring between urban planning and urban geography scholars, perhaps because the scoring criteria are more intuitive, as mentioned earlier scholars score 0 to 2 based on the presence or absence of mention, detailed discussion, and description. There are only a few indicators where scholars differ slightly in their scoring, for example, “% of energy from fossil fuels (natural gas, gasoline etc.)/hydro or other water intensive energy sources” “% of hardscape with surface reflectance index (SRI)>30”“ Does the plan address climate change adaptation?”“ Does the approach to flooding consider accommodate?”“Is the integration of adaptation and mitigation acknowledged?” and other criterion, urban geographers tended to score 2 points for specific data, while some urban planning scholars were more concerned with the level of detail in the engineering category of strategies to judge the scores. Overall, no significant differences were found between PhDs and Masters in the scoring for this assessment framework. This also indicates the general applicability and operability of the assessment framework.”
- We apologize for not uploading the appendix at the last round of submissions. Thank you very much for the reminder. In response to your comments, we have added the appendix after the references. Please see the response to comment 1 for a detailed appendix.
- In response to your comments, we have added the description of the integrated assessment in the first paragraph of our discussion. The modifications to this comment are combined with the modifications to comment 5, describing the corresponding indicator criteria, and adding the appendix after the reference. The appendix has the description of the indicators and criteria. The details are as follows.
“The Arden Structure Plan was published in July 2022. Although we did not assess further planning options, we did assess the planning issues, planning objectives, and strategies that the plan could address [55,56]; thus, this work is similar to an ex ante evaluation. A considerable effort was made to try to predict whether the planning objectives and solutions outlined in the Arden Structure Plan are scientific and relevant in the context of climate change and sustainable urban development. In this study, we integrated an assessment of urban sustainability and climate change with an assessment of the quality of the plan, as well as constructed an assessment framework that combined the quantitative and qualitative assessment of not only the urban sustainability and climate change elements of the Arden Structure Plan but also the feasibility and public participation of the plan. As Baer pointed out, there is no single standard for planning assessment; the established standards only provide a reference for planning assessment practitioners and are not decisive [57]. It is often possible to interpret them systematically by setting corresponding criteria [58]. For example, we have set up indicators such as "Implementation Section" and "Monitoring and Evaluation Section", through“Does the plan include a separate section that addresses what needs to be done to implement the plan?” and “Does the plan include a separate section that addresses what needs to be done to monitor and evaluate the plan?” and other criteria to assess the feasibility of the plan.;Set indicators for “Stakeholder and public engagement”, Public participation is assessed through the criteria "Does the plan identify the organizations and stakeholders involved in the plan making process and identify the public as part of the plan making process".”
Please see the response to comment 1 for the appendix.
- In response to your comments, we have added the corresponding descriptions in conjunction with comment 4 and added an appendix after the reference. The appendix has a description of the indicators and criteria. The details are as follows.
“The Arden Structure Plan was published in July 2022. Although we did not assess further planning options, we did assess the planning issues, planning objectives, and strategies that the plan could address [55,56]; thus, this work is similar to an ex ante evaluation. A considerable effort was made to try to predict whether the planning objectives and solutions outlined in the Arden Structure Plan are scientific and relevant in the context of climate change and sustainable urban development. In this study, we integrated an assessment of urban sustainability and climate change with an assessment of the quality of the plan, as well as constructed an assessment framework that combined the quantitative and qualitative assessment of not only the urban sustainability and climate change elements of the Arden Structure Plan but also the feasibility and public participation of the plan. As Baer pointed out, there is no single standard for planning assessment; the established standards only provide a reference for planning assessment practitioners and are not decisive [57]. It is often possible to interpret them systematically by setting corresponding criteria [58]. For example, we have set up indicators such as "Implementation Section" and "Monitoring and Evaluation Section", through“Does the plan include a separate section that addresses what needs to be done to implement the plan?” and “Does the plan include a separate section that addresses what needs to be done to monitor and evaluate the plan?”and other criteria to assess the feasibility of the plan.;Set indicators for “Stakeholder and public engagement”, Public participation is assessed through the criteria "Does the plan identify the organizations and stakeholders involved in the plan making process and identify the public as part of the plan making process". The assessment framework draws on relevant indicators from existing references [24,50] and adapts them to the water-sensitive geographical characteristics of Arden’s floodplain, focusing more on the impact of flooding on the area by setting indicators such as “groundwater pumping”, “percentage of 1-year, 24-h storm runoff for the entire community”, and “distance to critical floodplain/100–1000 years)”. The assessment framework is friendly and flexible for planners in water-sensitive areas and helps planners and decision makers to assess the quality of integrated urban sustainability and climate change plans. It is worth noting that there is no significant difference in scoring between urban planning and urban geography scholars, perhaps because the scoring criteria are more intuitive, as mentioned earlier scholars score 0 to 2 based on the presence or absence of mention, detailed discussion, and description. There are only a few indicators where scholars differ slightly in their scoring, for example, “% of energy from fossil fuels (natural gas, gasoline etc.)/hydro or other water intensive energy sources” “% of hardscape with surface reflectance index (SRI)>30”“ Does the plan address climate change adaptation?”“ Does the approach to flooding consider accommodate?” “Is the integration of adaptation and mitigation acknowledged?” and other criterion, urban geographers tended to score 2 points for specific data, while some urban planning scholars were more concerned with the level of detail in the engineering category of strategies to judge the scores. Overall, no significant differences were found between PhDs and Masters in the scoring for this assessment framework. This also indicates the general applicability and operability of the assessment framework.”
Once again, thank you very much for your comments and suggestions. And we hope that the corrections will meet with approval. If further revision is necessary, please contact us.
Thank you and best regards.
The authors.

Reviewer 3 Report
The revised version of the manuscript provided resonable responses to my concerns, whereas there are still two major problems need to be resolved. Firstly, the assessment framework is not clear, the data collection is not specified. Secondly, the figures is not clear and need to be improved to fulfill the requirement of IJERPH.
Author Response
Dear Editor Chen and reviewers,
Thank you for your letter and the reviewers’ comments on our manuscript entitled " The Effectiveness of Local Governments’ Policies in Response to Climate Change: An Evaluation of Structure Planning in Arden, Melbourne " (ID: ijerph-2123020). Those comments are very helpful for revising and improving our paper. We have studied the comments carefully and made corrections which we hope meet with approval. The main corrections are in the manuscript and the responds to the reviewers’ comments are as follows (the replies are highlighted in blue).
Reviewer 3:
The revised version of the manuscript provided resonable responses to my concerns, whereas there are still two major problems need to be resolved. Firstly, the assessment framework is not clear, the data collection is not specified. Secondly, the figures is not clear and need to be improved to fulfill the requirement of IJERPH.
Response:
Thank you for your comments. We have made the following modifications.
- We apologize for not uploading the appendix at the last round of submissions. Thank you very much for the reminder. In response to your comments, we have added the Plan quality assessment framework include assessment indicators and criteria description after the references. We describe the scoring rules in more detail in the scoring protocol section. The modifications are as follows.
“In order to assess the appropriateness of the sustainability and climate change con-siderations outlined in Arden’s plan and the feasibility of its actions, this study conducted a plan quality evaluation by assessing the content of the plan with respect to sustainability and climate change. Here, the AAA approach was adapted to form a new plan quality assessment protocol composed of 14 indicators designed focusing on these two themes. The overall framework is illustrated in Figure 2.”
“Appendix 1
Plan quality assessment framework |
||||||
Factors |
Category |
Indicator |
Objectives |
Description/Criteria |
Score |
Comments |
Awareness |
Context |
Context of Urban Sustainability and Climate change |
—— |
Does the plan include urban sustainability and climate change issues in the context of the site? |
1 |
The plan mentions the history, current development and shortcomings of Arden in the context, partly in relation to urban sustainability, but is vague. |
Concept |
Concept of Urban Sustainability and Climate change |
—— |
Does the plan include a description of the causes of climate change and the knowledge of urban sustainability? |
1 |
This plan does not specifically describe sustainability and climate change, but the strategy covers both two perspectives. |
|
Impacts |
Present impacts of Urban Sustainability and Climate change |
—— |
Does the plan include a discussion of the general impacts of climate change and urban sustainability? |
1 |
The plan mentions relevant content but lacks systematic descriptions. |
|
Targets |
Targets of Urban Sustainability and Climate change |
—— |
Does the plan include at least one goal related to climate change and urban sustainability? |
2 |
The plan proposes to embed sustainable change and some strategies about climate change. |
|
Analysis |
Urban Sustainability |
Environmental limits |
Natural Land Protection |
Nearby drinking water supply or source |
0 |
Arden is considering for an alternative water supply for indoor and outdoor non‐potable uses in Arden. Alternative water sources under investigation include local storm water harvesting and sewer mining. |
Nearby wastewater system |
0 |
The plan only Mentions the natural water management. |
||||
Nearby transportation infrastructure |
2 |
The plan propose high capacity public transport capable options and transport links connecting. |
||||
Brownfield area |
1 |
The plan mentions the management of land contamination |
||||
Existence of imperiled species |
0 |
This criterion is not mentioned, but should be considered. |
||||
Distance from wetlands (ft.) |
1 |
The plan describes that Arden is formerly a low‐lying wetland. |
||||
Distance from water bodies (ft.) |
1 |
Shown in Vision plan. |
||||
% of area (more than 15% slope) protected |
0 |
This criterion is not mentioned, but should be considered. |
||||
Residential dwelling units per acre of buildable land |
0 |
This criterion is not mentioned, but should be considered. |
||||
Use corridors etc. to form networks with the surroundings (Continuity of green space through rows of trees, shrubbery etc.). |
1 |
The plan proposes that green links and green streets will be continue to transform through extensive tree planting. |
||||
Environment building, considering natural flora in the surroundings. |
0 |
This criterion is not mentioned, but should be considered. |
||||
% of urban area protected as agriculture area for fresh food |
0 |
This criterion is not mentioned, but should be considered. |
||||
Ground water pumping |
2 |
Arden currently has five existing pumps and the Arden Flood Management Strategy proposes to upgrade the pumping stations and pipelines.The URCRS will also collect funds from developers to build pump stations. |
||||
% of 1 year, 24 hour storm runoff volume across the community |
0 |
No information in the plan, only a maximum storage capacity of 40 ML is mentioned. |
||||
% paved area in the designated area |
1 |
Objective 17 mentions that pedestrian priority areas will be enhanced by textured pavement changes, but there is no specific percentage. |
||||
Reuse of construction site soil |
2 |
Objective 3 and 6 refer to the adaptive reuse of built heritage sites to adapt to changing community needs |
||||
Combine utility lines to minimize land disturbance |
1 |
The plan proposes combining residential development with commercial, which means that some use routes will be combined, but this is not explicitly stated. Objective 4 proposes combining some blocks according to use needs, but still ensuring a connected, safe road network for smaller streets and driveways. |
||||
Consideration of scenic appearance |
2 |
Arden will be shaped by typical architectural forms. the architectural form of Laurens Streets complements and builds on the character of North Melbourne and will create a visual transition. |
||||
% of buildable land protected as greenspace |
0 |
This criterion is not mentioned, but should be considered. |
||||
Resource Efficiency |
% of boundary connected or adjacent to developed urban area |
1 |
The plan mentions the connection of boundary road with other North Melbourne areas. |
|||
% of Single family dwellings |
0 |
This criterion is not mentioned, but should be considered. |
||||
% of building sq. footage for which superstructure construction is employing BMPs1 for durability and moisture management |
0 |
This criterion is not mentioned, but should be considered. |
||||
% of building sq. footage for which substructure is employing BMPs for durability and moisture control |
0 |
This criterion is not mentioned, but should be considered. |
||||
% of building water use reduction below baseline buildings through the use of water efficient fixtures |
1 |
In the plan, water reuse and efficient energy consumption are embedded in the building design, but no specific data is provided. |
||||
Mutual use of centralized rainwater storage tanks |
2 |
Objective 20 mentions that new development can manage stormwater runoff from centralised wetlands through the provision of rainwater tanks. |
||||
% of landscape area fulfilling the criteria for efficient irrigation system |
2 |
Objective 14 proposes a minimum tree canopy cover of 40% by 2040 under Melbourne's Urban Forest Strategy. |
||||
% of wastewater reused |
1 |
Arden Vision 2018 only mentions the reuse of greywater & stromwater, percentage data is missing. |
||||
% capacity of alternative treatment system beyond the initial design |
1 |
Objective 10 refers to the early provision of alternative waste infrastructure on a regional scale to achieve sustainable development, but no specific percentage figures are available. |
||||
Considerations to minimize water leakage in water supply systems |
0 |
Information is lacking in the plan, but this is an important element to be considered. |
||||
% of Building sq. footage employing the use of energy efficient fixtures and technologies |
1 |
Objective 12 refers to ensuring that new technologies are introduced into buildings, such as energy trading technology and electrification of homes, but there is no data on percentages. |
||||
% of energy from fossil fuels (natural gas, gasoline etc.)/hydro or other water intensive energy sources |
1 |
Stategyc10.4 refers to the provision of fossil fuel free infrastructure for most areas, supporting net-zero carbon emission ambitions, but lacks basic data on the high energy consumption still in use. |
||||
% of energy from renewable sources |
2 |
Strategy 10.1 sets out the ambition for the region to have access to 100% renewable energy and to explore procurement opportunities to address the energy needs of the surrounding communities. |
||||
Centralized storage facilities for waste collection, facilities to reduce volume and employ composting |
2 |
Strategy 10.2 refers to promoting and encouraging the centralisation and sharing of waste management sites in order to reduce freight and waste vehicle emissions. |
||||
Use of locally-produced materials for building cladding, paving and other materials. |
2 |
Strategy 13.3 proposes to encourage the use of locally sourced building materials. |
||||
Reducing the generation of construction by-products |
2 |
Strategy 13.3 proposes to encourage the use of low carbon content feedstocks or production, where possible. |
||||
Sorting and recycling of construction by-products |
2 |
Strategy 13.3 proposes to encourage composting, recycling and reuse of construction materials. |
||||
Use of recycled products (recycled aggregate, blast furnace cement, electric furnace steel etc.) |
2 |
Strategy 13.3 proposes to encourage the use of renewable raw materials. |
||||
Use of materials with low climate change impact (blended cement etc.) |
2 |
Strategy 13.3 proposes to encourage the use of non-hazardous building materials. |
||||
Use of timber from sustainable forestry |
1 |
Strategy 13.3 indirectly expresses the use of sustainable materials, but does not mention sustainable forestry. |
||||
Environmental Quality |
Urban form that secures continuity in airflow within open spaces |
2 |
The plan mentions to design to ameliorates the effects of unsafe wind conditions, and deliver comfortable wind conditions in the public realm for walking, sitting or standing. |
|||
Counter measures against strong winds for regional characteristics |
0 |
The plan mentions wind control but no measures and seems not related to strong wind. |
||||
% of hardscape with surface reflectance index (SRI)>30 |
1 |
The plan requires all new buildings to use materials that minimise the urban heat island effect with a standard that at least 75 per cent of total project site areas should comprise of building or landscaping elements that increase the solar reflectance of the site. |
||||
% of building sq. footage for which BMPs are followed to ensure better indoor environmental quality |
0 |
This criterion is not mentioned, but should be considered. |
||||
Horizontal shaded area ratio from medium and tall trees, e.g., piloti, eaves, pergolas etc. |
2 |
The plan proposes that a target of 40 per cent canopy coverage should be achieved in the public realm over time. |
||||
% of residential area within 300 m walk from green space (>= 2 hectares in size) |
1 |
The service distance of public space is considered in the plan, but there is no specific data about residential area. |
||||
% of Building sq. footage designed for high performance building envelope |
2 |
Strategy 14.2 requires that new buildings reduce materials that lead to the urban heat island effect and that at least 75% of the buildings or landscape elements on the site should have increased solar reflectance. |
||||
Reduction of waste heat |
2 |
Objective 14 proposes that Melbourne is actively tackling this issue by 'greening' the public realm, cooling the environment by providing shade and transferring heat in the landscape. |
||||
% decrease in impervious area |
1 |
The urban water cycle & integrated water management mentions that in Arden, impervious surfaces have significantly altered local water systems and reduced evaporation, which Melbourne will mitigate through greening of public areas, but no specific data is available. |
||||
% of project pavements built with minimum albedo of 0.3 or permeable pavements/pavers |
0 |
This criterion is not mentioned, but should be considered. |
||||
% of streets with trees on both sides with avg. of 40 ft. spacing |
1 |
Strategy 22.3 proposes a tree cover of 40%, but does not address average spacing. |
||||
Distance from critical flood plain (100-1000 years) |
0 |
This criterion is not mentioned, but should be considered. |
||||
% of green space area to reduce heat island effect |
2 |
Th plan require all new buildings to meet a standard of 40 per cent total surface area as green cover; achieves a minimum canopy cover of 40 per cent by 2040. |
||||
Human Needs |
Public Mobility and Accessibility |
% of residential area within 600 m of stores, banks and administrative buildings |
1 |
There is a land function zoning map in the plan, but no specific data. Sub-precincts |
||
% of residential area within 600 m of medical and welfare facilities |
1 |
There is a land function zoning map in the plan, but no specific data. Sub-precincts |
||||
% of residential area within 600 m of educational and cultural facilities |
1 |
There is a land function zoning map in the plan, but no specific data. Sub-precincts |
||||
% of buildings frontage with 25 ft. or less setback |
1 |
Strategy 5.1 proposes controls for setbacks, but no specific data are available. |
||||
Measures taken to manage the traffic demand |
2 |
The plan proposes that Arden will prioritise people walking, cycling and using public transport to meet their daily needs. A network of walkable streets and protected cycle paths is planned. |
||||
Measures taken to calm traffic in urban centers |
2 |
The plan describes that Arden Central-Innovation to be activated by new metro station. |
||||
% of parking space reduction below local minimum or ECC criteria |
1 |
Objective 18 refers to controlling the number of parking spaces and encouraging public transport through parking overlays and other planning controls. However, specific data is missing. |
||||
Provide adequate bike lanes for roads with speed limit 35 mph or over |
1 |
Objective 16 refers to the provision of safe, direct and connected protected cycle paths. |
||||
Accommodation of bicycles near all primary building entrances |
0 |
Objecctive 17 proposes that the streets of the new station in Arden will be accessible to slow-moving cyclists, but does not specifically mention the entrances and exits of all major buildings. |
||||
% of roads (speed limit> 30 mph) with footpath width more than or equal to 900 mm on both sides |
0 |
This criterion is not mentioned, but should be considered. |
||||
% of bus-stop shelters with all-weather protection (rain, wind, snow) and renewable lighting/heating |
0 |
This criterion is not mentioned, but should be considered. |
||||
Visibility from the surroundings |
2 |
Objective 18 propose that new and existing streets will be pedestrian friendly and provide comfortable, green links between open spaces and public transport routes and enhance the quality of the public realm, so the urban spaces are visible and accessible. |
||||
Ease of approach by people with criminal intent |
0 |
There is no description in the plan about criminal intent. |
||||
Provision of street furniture, signs etc. |
1 |
The plan provide a providing a complete list of infrastructure items, but it does not mention street facilities in detail |
||||
Other considerations (lighting, sound etc.) |
2 |
Objective 28 popose to ensure that urban environmet is not unduly impacted by noise, vibration and electromagnetic interference from the adjacent railway corridor, elevated roadway and Metro Tunnel. |
||||
Provision of road width (at least 8m) and formation of a network of evacuation routes, in 2 directions etc. |
2 |
Objective 4 propose to create a new urban structure for Arden that incorporates a high quality network of connected streets, which has minimum 8m-width lanes. |
||||
Continuation of pre-existing community |
2 |
The plan is about urban regeneration, so it is a optimization of pre-existing comuunity |
||||
Social Well-being |
% of project pavements that reduce tire pavement noise levels below certain limits |
0 |
This criterion is not mentioned, but should be considered. |
|||
Noise and vibration counter measures |
2 |
Objective 28 refers to the state government's efforts to relocate industrial businesses out of the district to ensure that noise disturbance from industrial sites is eliminated. The tunnel design of the metro also includes measures to mitigate vibration and noise disturbance. |
||||
Provision of crime prevention facilities, such as surveillance cameras and guards (surveillance of blind spots) |
2 |
Strategy 17.5 proposes through the use of Crime Prevention Through Environment Design principles to promote the use of open space and the ground floor around major pedestrian routes to maximise personal safety. |
||||
Economic Prosperity |
% of regularly trafficked lanes designed for long-life (>=40 years) |
0 |
No information is mentioned in the plan. |
|||
Efforts to track pavement performance through testing and condition assessment procedures |
1 |
Strategy 17.7 proposes to review and update road speed limits in accordance with Arden's principles for movement, but there are no specific procedures or measures. |
||||
Development of facilities related to local industries and culture |
2 |
Objective1propose to celebrate, protect and interpret Aboriginal cultural values and heritage in the planning, design and curation of Arden |
||||
% of labor from local area |
2 |
Strategy 3.1 propose that facilitate use and development of land in Arden to deliver on the aspiration of approximately 34,000 jobs and around 15,000 residents in the precinct. |
||||
Proportion of total house and land packages delivered to the market at an affordable purchase price (less than 270,000$ - 2008 figure - Australia) for moderate income households. |
2 |
Oblective 23 propose that at least 6% of all new housing in Arden is affordable for very low to moderate income households and delivered as social and affordable housing or shared equity. |
||||
% of land costed at lowest quartile of local market |
0 |
There is nothing in the plan about land costs. |
||||
Social Equity |
Public Mobility and Accessibility |
Realization of barrier free outside spaces for the infirm or handicapped. |
1 |
Strategy 18.2 plan to provide disabled parking. Strategy 25.5 propose to provide diverse and adaptable community facilities to become benificial to people with disablities, but the measures are quite general. |
||
Participation by residents of the designated area in planning processes |
0 |
There is nothing in the plan about the participation by residents. |
||||
Consideration of encounter |
2 |
Objective 21 propose to provide generous, well-designed and accessible open spaces that are diverse and flexible to meet the needs of Arden’s evolving community and visitors to the precinct. |
||||
% of primary entryways facing public spaces |
0 |
This criterion is not mentioned, but should be considered. |
||||
Resource Efficiency |
Reasonable redistribution of resources for vulnerable groups |
1 |
Objective 5 propose to make Arden become a world leading urban renewal and innovation precinct. Urban renewal is one of a way for resources redistribution. |
|||
Social Well-being |
Embrace and recognize differences |
0 |
The plan does not mention the recognition and inclusion of social differences |
|||
Overcome gentrification and residential segregation |
2 |
Objective 23 propose to facilitate inclusive, well-designed, sustainable and accessible housing, with affordable housing for very low to moderate income households and delivered as social and affordable housing or shared equity. |
||||
Climate change |
Mitigation |
—— |
Does the plan address climate change mitigation? |
2 |
The plan promotes climate change mitigation, such as minimising waste production and mitigating the urban heat island effect. |
|
Is a mitigation aim specified(e.g. specific level of warming limit, goals for GHG emission reducetion?) If yes, what level? |
2 |
Objective 9 propose to establish strong environmental governance in Arden to achieve the precinct’s net-zero carbon emissions target by 2040, which is adequate to achieve Pairs Argreement. |
||||
Affordable housing mandates |
2 |
Oblective 23 propose that at least 6% of all new housing in Arden is affordable for very low to moderate income households and delivered as social and affordable housing or shared equity. |
||||
Mixed use zoning to support self-contained communities |
2 |
Obljective 25 propose to ensure timely delivery of high-quality, accessible and integrated community infrastructure to meet the needs of existing and future residents, workers and visitors. |
||||
Transfer of development rights to support transit-oriented conrridors |
1 |
Objective 15 propose to provide space for high capacity public transport capable options and improving transport links connecting Arden into the expanding central city, but it doesn's mention the development rights. |
||||
Mixed use zoning to encourage urban regeneration and infill |
2 |
The plan focuses on the urban renewal of Arden, and encourage efficient and diverse use of land. |
||||
Zoning overlays and form-based codes to achieve desired outcomes such as new urbanism |
2 |
Strategy 5.1 implement built form controls in the planning scheme that respond to key design recommendations. |
||||
Design codes and flexible parking to achive transit-oriented development |
2 |
Strategy 18.1 propose to prepare planning controls that direct the ongoing supply and location of car parking to achieve the 10% mode share target for private vehicles and car parking principles, and then minimise the impact of car parking and associated vehicular movements through Arden. |
||||
Mixed-use zoning to support eco-communities |
1 |
The plan concentrates on the sastainable change in community,including some mesures of eco-communities, but it dose not have a systematic description. |
||||
Code revisions/design guidelines for pedestrian zones |
2 |
Strategy 17.1 propose to facilitate a network of permeable streets and pedestrian links through the precinct for walking and cycling that are considerate of safety and convenience and provide direct access to and from key destinations. |
||||
Code revisions/design guidelines to achive car-free disctricts &/or traffic calming |
1 |
Strategy 17.1 encourage the consolidation of servicing facilities and alternative freight delivery models within the precinct to reduce the number of vehicles entering and circulating, but there is no description of car-free disctrict. |
||||
Adaptation |
—— |
Does the plan address climate change adaptation? |
2 |
Objective 12 propose to measure the performance of the precinct, its buildings and its occupants and be able to adapt to changes in climate, lifestyle and technology in the future. |
||
What level of flooding risk is planned for, and across what time frame? |
0 |
The plan mentions flooding management, but there was no clear planning level or timing. |
||||
Does the approach to flooding consider retreat? |
0 |
The plan does not consider retreat for flooding |
||||
Does the approach to flooding consider accommodate? |
2 |
Strategy 19.4 positively respond to any necessary level changes that are required for drainage purposes between development and the public realm, to accommodate flooding problems. |
||||
Does the approach to flooding consier protect? |
2 |
There is the precinct-wide flood management strategy for Arden, to provide guidance on how development can achieve flood responsive design and good urban design outcomes that facilitates safety, equitable access and universal design. |
||||
Integration of mitigation and Adaptation |
—— |
Is the integration of adaptation and mitigation acknowledged? |
1 |
The plan mentions both adaptation and mitigation, but neither is called one-sided and is not described systematically. |
||
Action |
Implementation |
Implementation Section |
—— |
Does the plan include a separate section that addresses what needs to be done to implement the plan? |
2 |
Section 10 called delivery Arden talks mention the development staging of Arden. |
Monitoring and evaluation |
Monitoring and Evaluation Section |
—— |
Does the plan include a separate section that addresses what needs to be done to monitor and evaluate the plan? |
1 |
The plan just mentions the lead agency. |
|
Inter-organizational coordination |
Vertical and horizontal integration |
—— |
Does the plan include at least one vertical connection to federal, provincial plans and regional plans or at least one horizontal connection with other local plans/programs? |
2 |
The plan has connection with other plans such as Moonee Ponds Creek Strategic Opportunities Plan. |
|
Participation |
Stakeholder and public engagement |
—— |
Does the plan identify the organizations and stakeholders involved in the plan making process and identify the public as part of the plan making process |
1 |
The plan just mentions the lead agency and lacks the public participation |
”
“The scholars scored the Arden structural plan with reference [52,53] to the scoring used in previous studies. The evaluation of the quality of the Arden Structure Plan was conducted by scoring the indicators on a scale between 0 and 2 [50]. A score of ‘0’ was given if there was no evidence or description of the indicator throughout the plan. An indicator was given a score of ‘1’ if the indicator was inferred or mentioned but not in detail, while an indicator was given a score of ‘2’ when it was fully mentioned and discussed in detail. Each factor (AAA) was equally important and had the same weight. The scoring protocol was implemented with comments on each indicator to justify the coding decision. The evaluation protocol contains 14 indicators with a maximum score of 28 points. These indicators are divided according to AAA as follows: awareness (4/14), analysis (6/14) and action (4/14). Scoring is done by category, with a maximum of 8 points for awareness, 12 points for analysis and 8 points for action. In addition, an assistant professor from the University of Melbourne was invited to validate the scoring protocol to ensure professionalism and reduce the influence of subjective factors by averaging the results.”
- In response to your comments, we redrew the figures according to the requirements of IJERPH. The modifications are as follows.
Figure 1. Arden Location Map
Figure 2. Plan quality assessment framework.
Figure 3. Fieldwork route and photos.
Figure 4. Arden’s flood area with an increase in sea level of 2.7 metres. Source: foreground, 2018, https://www.foreground.com.au/planning-policy/reimagining-australias-temperate-kakadu/ (accessed on 20 September 2022).
Figure 5. Analysis of core factors.
Figure 6. Analysis of environmental limits.
Figure 7. Analysis of human needs.
Figure 8. Analysis of social equity.
Figure 9. Analysis of climate change evaluation.
Once again, thank you very much for your comments and suggestions. And we hope that the corrections will meet with approval. If further revision is necessary, please contact us.
Thank you and best regards.
The authors.
